# Characteristics of Physicochemical Properties of Chalky Grains of *Japonica* Rice Generated by High Temperature during Ripening

**DOI:** 10.3390/foods11010097

**Published:** 2021-12-30

**Authors:** Sumiko Nakamura, Ayaka Satoh, Masaki Aizawa, Ken’ichi Ohtsubo

**Affiliations:** Faculty of Applied Life Sciences, Niigata University of Pharmacy and Applied Life Sciences, 265-1 Higashijima, Akiha-ku, Niigata 956-8603, Japan; snaka@nupals.ac.jp (S.N.); ayaka.s981027@gmail.com (A.S.); masa.301512@gmail.com (M.A.)

**Keywords:** rice, high-temperature damage, retrogradation degree of hardness, α-amylase activity

## Abstract

Global warming has caused devastating damage to starch biosynthesis, which has led to the increase in chalky grains of rice. This study was conducted to characterize the qualities of chalky rice grains and to develop the estimation formulae for their quality damage degree. We evaluated the chalkiness of 40 *Japonica* rice samples harvested in 2019, in Japan. Seven samples with a high ratio of chalky rice grains were selected and divided into two groups (whole grain and chalky grain). As a results of the various physicochemical measurements, it was shown that the surface layer hardness (H1) of cooked rice grains from chalky grains was significantly lower, and their overall hardness was significantly lower than those from the whole grains. In addition, α- and β-amylase activities, and sugar contents of the chalky rice grains were significantly higher than those of the whole rice grains. The developed estimation formula for the degree of retrogradation of H1 based on the *α*-amylase activities and pasting properties, showed correlation coefficients of 0.84 and 0.81 in the calibration and validation tests, respectively. This result presents the formula that could be used to estimate and to characterize the cooking properties of the rice samples ripened under high temperature.

## 1. Introduction

Rice (*Oryza sativa* L.) is one of the main food crops throughout the world and the staple food for about half of the global population [1]. Rice is used as table food and for various food processing applications such as rice snacks, rice cakes, and sake wine, etc.

As starch is the main component of rice grains, its characteristics, such as apparent amylose content (AAC), amylopectin chain length distribution, and resistant starch, affects rice quality markedly.

Rice quality consists of apparent quality and internal quality. The former quality is evaluated through a human test by trained inspectors and graded based upon the ratio of whole grains, damaged grains including chalky grains, foreign grains and foreign matters, and grain characteristics. The latter quality means the eating quality (palatability), processing suitability, etc. and is evaluated by the sensory test, experimental processing test, or the various kinds of physicochemical measurements [2,3].

The quality evaluation of rice is carried out by the application of a sensory test and physicochemical measurements.

Among the physicochemical measurements, analysis of the pasting properties of rice starch by a Rapid Visco Analyzer (RVA) and the texture of the cooked rice grains are very important. The pasting property and texture are affected by the starch properties because starch is a main component of rice grains.

We previously found that the iodine absorption curve of rice starch differed among the various sample rice cultivars and we developed the estimation formulae for AAC, resistant starch (RS) content, and amylopectin chain length distribution [4,5]. We performed an analysis of the pasting properties using an RVA and developed the novel estimation formulae for AAC and RS [6], and for surface hardness after retrogradation [7]. Furthermore, we developed novel estimation formulae for oleic and linoleic acid contents based on the pasting properties of brown rice flours measured by an RVA [8].

According to the Sixth Assessment Report of the Intergovernmental Panel on Climate Change (IPCC), the global surface temperature in the first two decades of the 21st century (2001–2020) was 0.99 °C higher than 1850–1900, and global warming of 1.5 °C and 2 °C will be exceeded during the 21st century unless deep reductions in CO_2_ and other greenhouse gas emissions occur in the coming decades [9].

It has been observed that recent global warming has caused remarkable damage in the quality and appearance of crops [10,11]. High-temperature stress during grain filling hastens the growth rate of endosperm and causes grain chalkiness [10]. As a result of high-temperature damage, grain weight, width, and thickness has decreased [12,13].

Furthermore, the ripening of rice grains under high temperatures tends to increase chalky grains, which leads to a lower milling yield and higher generation of cracked grains during the rice milling. Therefore, the Ministry of Agriculture, Forestry and Fisheries, Japan, issued a National Inspection Standard in which chalky grain rich rice is ranked as a lower grade.

As low-grade rice sells at a lower price in the market, the generation of chalky rice grains causes a severe problem not only to the farmers but also the milling companies, wholesalers, and retailers. As chalky grains lead to the low palatability of the cooked rice, this also bring about demerits to the consumers.

Mitsui et al. [14] showed that high-temperature stress revealed the downregulation of genes for starch synthesis enzymes and the upregulation of genes for *α*-amylases. Asaoka et al. [15] showed that a higher environment temperature during the ripening of rice grains decreases the amylose content in endosperm starches of *Japonica* rice cultivars. Tsutsui et al. reported that unusual starch degradation by α-amylase is involved in the occurrence of chalky grains rather than starch synthesis [16].

Formerly, many researchers reported the damage to rice grains by high temperatures [17,18]. Lanning et al. [19] reported that extreme nighttime air temperatures impact rice chalkiness and milling quality. Nevertheless, most of those reports were the result of a focus on the yield and apparent quality, such as the decrease in the yield of rice [10,13,20], generation of chalky rice grains [21], elucidation of the mechanism of high-temperature damage [22], breeding of high temperature-tolerant rice cultivars [23], and the relationship between high-temperature damage and the activation of α-amylase [16], in which all of the total rice grains were used as samples.

Patindol and Wang investigated the fine structures and physicochemical properties of starches from chalky and translucent rice kernels [24]. They reported that the starch from the chalky kernels contained less amylose and more short-chain amylopectin. Singh et al. [25] separated chalky grains and translucent grains from three rice cultivars, and reported morphological, thermal, cooking, and textural properties of the two groups. Chun et al. [26] compared the chalky rice and head rice in terms of chemical composition, morphological structure, cooking, the textural properties of cooked rice, and pasting properties.

High-temperature damage of rice starch reflects the deterioration in eating properties, such as hardness, stickiness, and pasting properties. Therefore, it is necessary to develop methods to evaluate or estimate the degree of quality damage caused by high temperature.

During storage, several physicochemical and physiological changes occur in rice foods including loss of moisture and aroma, ultimately imparting hardness and cracking, technically referred to as retrogradation [27].

Recently, the consumption of rice at home has been decreasing, and the consumption of rice from take-out lunch-boxes and rice dishes served at restaurants has been increasing in Japan, Korea, and China [28]. In such cases, the quality of the cooked rice is evaluated not only just after cooking but also several hours after cooking. Therefore, the degree of retrogradation has become an important quality index for rice and rice products [7]. Yang et al. reported that the addition of emulsifier could decrease the gelatinization enthalpy (Δ𝐻𝑔), retrogradation index, and hardness and increase the adhesiveness and water content [29].

In this study, we tried to clarify how the quality of chalky rice grains was damaged by high temperatures in 2019 when it was extraordinary hot during the summer all over Japan. After sorting a typical sample of rice grains, we evaluated and compared the two groups of rice grains, a whole rice group and chalky rice group, using various physicochemical measurements. Furthermore, we tried to develop an estimation formulae for the degree of quality damage of chalky rice grains ripened under high temperatures in 2019.

## 2. Materials and Methods

### 2.1. Materials

The rice samples were purchased in 2019 at a local market and were subjected to the measurement in 2020 (*Japonica* subspecies) (*n* = 40). Seven rice samples (shown in bold letters below) with a high proportion of chalky grains were selected from 40 *Japonica* rice samples. Tsuyahime (cultivated in Shimane), *Sagabiyori* (Saga), *Oidemai* (Kagawa), *Hatsushimo* (Gifu), *Akihonami* (Kagoshima), *Nikomaru* (Kouchi), *Koshihikari* (Hyogo), *Morinokumasan* (Kumamoto), *Yumeshizuku* (Saga), *Ichihomare* (Fukui), *Hinohikari* (Kouchi), *Ryunohitomi* (Gifu), *Koshihikari* (Niigata; Muikamachi), *Koshihikari* (Niigata; Shibata), *Koshihikari* (Niigata; Myokou), *Kumasannochikara* (Kumamoto), *Shinnosuke* (Niigata), *Kinumusume* (Shimane), *Sainokizuna* (Saitama), *Tentakaku* (Toyama), *Hanaechizen* (Fukui), *Fusaotome* (Chiba), *Koshihikari* (Niigata; Oguni), *Koshihikari* (Niigata; Yamakoshi), *Koshihikari* (Niigata), *Koshihikari* (Nagano), *Tsuyahime* (Yamagata), *Hitomebore* (Miyagi), *Nanatsuboshi* (Hokkaido), *Yumepirika* (Hokkaido), *Tennotsubu* (Fukushima), *Sasanishiki* (Miyagi), *Akitakomachi* (Akita), *Seitennohekireki* (Aomori), *Ginganoshizuku* (Iwate), *Yukiwakamaru* (Yamagata), *Datemasayume* (Miyagi), *Koshihikari* (Niigata; Tochio), *Koshihikari* (Niigata; Toukamachi), *Koshihikari* (Niigata; Sado). Each samples was stored at 10 °C in rice storage chamber.

### 2.2. Ratios of Whole and Chalky Rice Grains

Milled rice grains were classified into whole and chalky rice grains using a grain discriminator (Grain Quality Inspector RGQ120; SATAKE, Corp., Higashihiroshima, Japan) and visually separated the whole and chalky rice grains from 7 rice samples.

### 2.3. Preparation of Milled Rice Flour

Milled rice flour was prepared from whole and chalky rice grains using a cyclone mill (SFC-S1; UDY, Corp., Fort Collins, Co, USA) with a screen containing 1-mm-diameter pores.

### 2.4. Preparation of Starch Granules

Starch granules were prepared from whole and chalky rice flour using the cold alkaline method [30,31].

### 2.5. Iodine Absorption Spectrum

The AACs of alkali-treated rice starch were measured using the iodine colorimetric method of Juliano [32]. The iodine absorption spectrum of alkali-treated whole and chalky rice grains starch was measured using a Shimadzu UV-1800 spectrophotometer. Peak wavelength on iodine staining of starch (λ_max_) shows high correlation with the length of glucan chain, molecular size of amylose and super-long chain (SLC) of amylopectin, and absorbance at λ_max_ (A_λmax_) [4,5]. “New λ_max_” is calculated as follows
New λ_max_ = (73.307 × A_λmax_ + 0.111 × λ_max_ − 73.016)/(λ_max_ of various rice starches − λ_max_ of glutinous starch)(1)

New λ_max_ shows positive high correlation with resistant starch [4,5].

### 2.6. Pasting Properties

The pasting properties of whole and chalky rice flours from 7 *Japonica* rice samples were measured using a Rapid Visco Analyzer (RVA) (model Super 4; Newport Scientific Pty, Warriewood, Australia). A program for heating and cooling cycle by Toyoshima et al. was adopted [33].

Novel indices such as the Set/Cons ratio and Max/Fin ratio are very strongly correlated with the proportion of intermediate and long chains of amylopectin: Fb_1+2+3_ (DP ≥ 13) [6].

### 2.7. Measurement of RS

The RS of whole and chalky cooked rice flours were measured according to the AOAC method using an RS assay kit (Megazyme, Ltd., Wicklow, Ireland) [7]. In detail, the cooked rice flours were prepared by pulverization after lyophilization, and each sample (100 mg) was digested with pancreatin and amyloglucosidase at 37 °C for 6 h. Finally, reaction product, glucose content, was measured using a spectrophotometer at 510 nm.

### 2.8. Physical Properties of Cooked Rice Grains

The hardness and stickiness of the cooked rice grains were measured using a Tensipresser (My Boy System, Taketomo Electric Co., Tokyo, Japan) with the individual grain method in low compression (25%) and high compression (90%) tests [34]. The average of each parameter was calculated by measuring 20 individual grains.

As a staling test for the cooked rice, the cooked samples were stored at 6 °C for 24 h and measured again with a Tensipresser according to the previously described method in low compression (25%) and high compression (90%) tests [35].

### 2.9. Measurement of Glucose and Sucrose Contents

The cooked rice flour sample was prepared by pulverization after lyophilization. The glucose and sucrose contents were measured by the enzyme assay method of NADPH using the glucose and sucrose assay kit (F-kit, Roche/R-Biopharm Darmstadt, Germany).

### 2.10. α-Amylase Activity

α-Amylase activity of the whole and chalky rice grains was determined by the enzyme kit (Megazyme International Ireland).

### 2.11. β-Amylase Activity

β-Amylase activity of the whole and chalky rice grains was determined by the enzyme kit (Megazyme International Ireland). For *β*-amylase activity measurement, rice flour (0.1 g) was extracted with 0.7 mL of extraction buffer, pH 8.0 at 20 °C for 60 min, and thereafter centrifuged for 15 min at 2000× *g*. Extraction solution (0.3 mL; 2-fold dilution) and substrate (0.1 mL) were preincubated at 40 °C for 5 min. Thereafter, each sample solution was incubated at 40 °C for exactly 10 min, followed by the addition of stopping reagent (3.0 mL). The absorbance was measured at 400 nm.

### 2.12. Sodium Dodecyl Sulfate–Polyacrylamide Gel Electrophoresis (SDS-PAGE)

Protein was extracted from the whole and chalky milled rice flour samples (0.5 g) by shaking with 2 mL of buffer A (50 mM Tris-HCl, pH 6.8, 2% SDS, 5% 2-mercaptoethanol) at 37 °C for 30 min, and then centrifuged for 5 min at 3000× *g*. The supernatant (1 mL) was diluted with an equal volume of sample buffer (0.125 M Tris-HCl pH6.8, 10% 2-mercaptoethanol, 4% SDS, 10% sucrose, 0.004% bromophenol blue) and mixed well before heating for 2 min at 100 °C. In total, 10 μg of extracted protein was loaded into each lane in 12% polyacrylamide gel [34]. The values were calculated based on the intensities of various spots on the gel after SDS-PAGE analysis using the ATTO densitograph software library (CS Analyzer ver 3.0).

### 2.13. Statistical Analyses

All the results, including the significance of regression coefficients, were statistically analyzed using Student’s *t*-test, one-way analysis of variance, and Tukey’s test with Excel Statistics (ver. 2006; Microsoft Corp., Tokyo, Japan).

## 3. Results and Discussion

### 3.1. Pasting Properties of 40 Rice Samples and Sorting of Chalky Rice Grains

The pasting properties of 40 *Japonica* rice samples are shown in Table 1. They showed versatile pasting properties reflecting the difference in cultivars, locations, and climate conditions. In our previous study [28], the pasting properties of 38 rice samples were measured using an RVA, by which we developed novel estimation formulae for estimating apparent amylose content (AAC) and resistant starch (RS). Using the same estimation formulae, we estimated AAC and RS values and these are shown in Figure 1. Usually, high-amylose rice tends to contain a high amount of RS. Nevertheless, in this study, several low-amylose rice samples showed high RS contents, and some intermediate-amylose rices revealed low RS contents, as shown in Figure 1. It seems that the temperature during ripening affected the AAC and RS of various rice samples.

As shown in Figure 2, the ratios of the chalky rice grains of seven rice samples (shown with a down arrow) (8.2–28.0%; mean, 17.6%) were higher than those of 33 *Japonica* rice samples (0–9.2%; mean, 2.9%), and the ratios of whole rice grains of seven rice samples (56.8–87.0%; mean, 70.9%) were lower than those of 33 *Japonica* rice samples (60.0–97.9%; mean, 88.5%). In particular, the Koshihikari (Niigata; Toukamachi), Koshihikari (Niigata; Oguni), and Koshihikari (Niigata; Tochio) samples showed a high ratio of chalky rice grains because those rice samples were damaged by high temperatures resulting from the foehn phenomenon due to a typhoon in 2019.

Appendix Aa shows the relationship between the ratios of chalky rice grains and estimated AACs, and Appendix Ab shows the relationship between the ratios of chalky rice grains and estimated RS contents. A dotted circle shows a high ratio of whole rice grains, so we did not select this one.

As a result, we selected seven rice samples with a high ratio of chalky rice grains from 40 *Japonica* rice samples in 2019. As shown in Appendix A, the environmental temperature for the selected seven rice samples during the rice grain ripening was very high compared with the temperature in the ordinary year.

### 3.2. Iodine Absorption Spectrum

Low-amylose rice generally becomes soft and sticky after cooking, whereas high-amylose rice becomes hard with fluffy separated grains [36]. The group of high-amylose rice starches includes two types of starch functions, which are amylose and super-long chains (SLC) of amylopectin [37]. Igarashi et al. [38] showed that the apparent amylose content (AAC) and the super-long chains (SLC) of amylopectin in Hokkaido varieties increased at lower grain-filling temperatures. The amylose content of rice is controlled mainly by the Wx gene GBSS (granule-bound starch synthase) [39,40]. The starches in the rice cultivars grown under low temperatures have significantly higher amylose content and lower SLC amylopectin content than cultivars grown under high temperatures [41,42]. Taira et al. [43] reported that the lipid content and fatty acid composition of rice were affected by daily mean temperature during ripening. Inouchi et al. [44]^.^ showed that the SLC content of starch can be estimated on the basis of λ_max_ and the blue value of purified amylopectin. Furthermore, Igarashi et al. [45] reported a positive correlation between absorbance at λ_max_ and AAC.

Table 2 shows that the absorbance values at 620 nm (representing AAC) of the starch of chalky rice grains (10.7–15.7%; mean, 13.5%) were lower than those of the starch of whole rice grains (11.9–16.5%; mean, 14.3%). Particularly, starch of the chalky rice of Morinokumasan, Koshihikari (Toukamachi), Hanaechizen, and Sasanishiki showed significantly lower AACs than those of the starch of whole rice grains. The AACs of the starch of chalky rice were 0.9 times lower than those of the starch of whole rice. These results are in accordance with those reported by Patindol and Wang [21] and Chun et al. [23], and mean that high temperatures cause shortening of the starch molecule.

The λ_max_ of the starch of chalky rice grains (563.0–572.5 nm; mean, 568.0 nm), were lower than those of the starch of whole rice grains (566.5–578.5 nm; mean, 572.0 nm). In particular, the λ_max_ of the starch of the chalky rice of Koshihikari (Shibata) was significantly lower than that of the starch of whole rice grains. The λ_max_ of the starch of chalky rice grains were slightly lower than those of the starch of whole rice grains. The λ_max_ of high molecular weight amyloses are generally longer and the λ_max_ values of glutinous rice cultivars are very low [26,27].

The λ_max_/AAC ratios of starch of chalky rice grains (36.1–53.0; mean, 43.0) were higher than those of the starch of whole rice grains (34.4–48.6; mean, 40.5). In particular, the starch of the chalky rice of Morinokumasan, Koshihikari (Toukamachi), and Sasanishiki were significantly higher than those of the starch of whole rice grains. The λ_max_/AAC ratios of the starch of chalky rice grains were 1.1 times higher than those of the starch of whole rice grains.

The Aλ_max_ values of ae mutant rice cultivars tend to be higher than those of other cultivars because the others contain SLC [45]. The Aλ_max_ of the starch of chalky rice grains (0.263–0.340; mean, 0.303) were lower than those of the starch of whole rice grains (0.26–0.348; mean, 0.312). In particular, the Aλ_max_ ratios of the starch of the chalky rice of Morinokumasan and Sasanishiki were significantly lower than those of the starch of whole rice grains. The Aλ_max_ of the starch of chalky rice grains were slightly lower than those of the starch of whole rice grains. These results are in agreements with the studies of Takeda et al. [37] and Igarashi et al. [38].

The λ_max_/A_λmax_ ratios of glutinous rice starches were higher than other cultivars [26,27]. The λ_max_/A_λmax_ ratios of the starch of chalky rice grains (1658.3–2161.9; mean, 1892.2) were higher than those of the starch of whole rice grains (1631.7–2150.6; mean, 1848.0). In particular, the λ_max_/A_λmax_ ratios of the starch of the chalky rice of Morinokumasan, Koshihikari (Toukamachi), Hanaechizen, and Sasanishiki were significantly higher than those of the starch of whole rice grains. The λ_max_/A_λmax_ ratios of the starch of chalky rice grains were slightly higher than those of the starch of whole rice grains. In a previous study, we showed that AAC was negatively correlated with the λ_max_/A λ_max_ ratio [5].

The Fb_3_ (DP ≥ 37) (the ratio of long chains of amylopectin) [4,5] of the starch of chalky rice grains (11.0–14.4%; mean, 12.8%) were lower than those of the starch of whole rice grains (11.2–14.8%; mean, 13.2%). In particular, the Fb_3_ (DP ≥ 37) of the starch of the chalky rice of Morinokumasan and Koshihikari (Toukamachi) were significantly lower than those of the starch of whole rice grains. The Fb_3_ (DP ≥ 37) of the starch of chalky rice grains were slightly lower than those of the starch of whole rice grains. The Fb_3_ (DP ≥ 37) of ae mutant rice were higher than those of other rice cultivars [41]. These results are in agreements with the reports by Inouchi et al. [44] and Igarashi et al. [45].

The “New λ_max_” value is assumed to be related to the SLC content of amylopectin [4,5]. The New λ_max_ values of the starch of chalky rice grains (0.241–0.390; mean, 0.302) showed a similar tendency with those of the starch of whole rice grains (0.215–0.375; mean, 0.296).

The molar ratio of short chains to long chains of the amylopectin fraction seemed to be one of the useful evaluation/selection indexes for breeding. The Fb_3_/Fa (Fa; short-chain fraction of amylopectin) of the starch of chalky rice grains (0.42–0.78; mean, 0.60) were lower than those of the starch of whole rice grains (0.43–0.80; mean, 0.63). In particular, the Fb_3_/Fa of the starch of the chalky rice of Morinokumasan and Koshihikari (Toukamachi) were significantly lower than those of the starch of whole rice grains. The Fb_3_/Fa ratios of the starch of chalky rice grains were slightly lower than those of the starch of whole rice grains. As a result, it seems that the starch properties of Morinokumasan, Koshihikari (Toukamachi), and Sasanishiki showed a marked tendency of lower AACs and higher λ_max_/AAC than those of the starch of whole rice grains.

Our results were consistent with those of Asaoka et al. [15] and Singh et al. [25], and Chun et al. [26] in terms of AAC; furthermore, we showed that not only AAC but also chain length distribution (Fb_3_/Fa) changed in the case of the chalky grains, and that λ_max_/AAC may be a good indicator for the degree of quality deterioration of the chalky grains.

### 3.3. Pasting Properties

Pasting properties are useful quality indicators because they affect the eating quality of rice [6]. The Fin. Vis. (Final viscosity) of high-amylose rice cultivars has been shown to be higher than those of low-amylose cultivars, and Fin. Vis. and Cons (consistency) are related to the degree of starch retrogradation during cooling [46,47]. In the previous study, we developed a novel index of the ratios of setback/consistency (SB/Cons) and (Max/Fin), which positively or negatively correlated with the proportion of intermediate and long chains of amylopectin (Fb_1+2+3_ (DP ≥ 13) [6]. Okuda et al. [48] showed that a high air temperature during one month after heading showed a high positive correlation with gelatinization temperatures for sake rice cultivars.

Table 3 shows that the Max. vis. (maximum viscosity) of chalky rice grains (320.9–420.7 RVU; mean, 372.4 RVU) were lower than those of whole rice grains (352.4–446.0 RVU; mean, 401.0 RVU). Moreover, the Mini. Vis. (minimum viscosity), BD (breakdown), Fin. Vis. and Cons (consistency; Fin. Vis.–Mini. Vis.) of almost all the chalky rice grains were significantly lower than those of the whole rice grains. The pasting properties of chalky rice grains were 0.9 times lower than those of whole rice grains, whereas the Pts (pasting temperatures) of chalky rice grains were slightly higher than those of whole rice grains. This result agreed with the results reported by Okuda et al. [46,47] and Chun et al. [26]. Moreover, the SB/Cons (setback/consistency), Max/Min (maximum viscosity/minimum viscosity), and Max/Fin (maximum viscosity/final) of chalky rice grains were almost the same as those of whole rice grains.

As a result, in all samples, the Mini. Vis. of chalky rice grains were significantly lower than those of whole rice grains and those of Max. vis. showed a similar tendency. It seems that the pasting properties of all samples were affected by high-temperature damage. In particular, the chalky rice grains showed a tendency for lower Max. vis., Mini. Vis. and Fin. vis. than those of whole rice grains.

Singh et al. [25] reported that the chalky grains had higher enthalpy on DSC. We showed that both the Max. vis. and Mini. Vis. were lower than those of whole grains in pasting properties using an RVA. This would likely be due to the higher activities of α- and β-amylases of the chalky grains.

### 3.4. Physical Properties of Cooked Rice Grains

The cooked rice samples were kept in the vessel at 25 °C for 2 h and then subjected to measurements. As shown in Table 4A, the H1 (hardness of surface layer) of chalky cooked rice grains (0.042–0.066 × 10^5^ (N/cm^2^); mean, 0.049 × 10^5^(N/cm^2^)) were significantly lower than those of overall cooked rice grains (0.054–0.067 × 10^5^ (N/cm^2^); mean, 0.063 × 10^5^ (N/cm^2^)) at *p* < 0.01, and those of H2 (hardness of overall layer) showed a similar tendency at *p* < 0.05. The H1 and H2 of chalky cooked rice grains were 0.8~0.9 times lower than those of overall cooked rice grains.

Singh et al. [25] reported that the chalky kernels from different varieties show lower values for cooking and textural parameters than the translucent kernels, and Chun et al. [26] reported that cooked chalky kernels became harder and less adhesive. Although the results differ according to the difference in rice samples and climate conditions, our results were consistent with the report by Singh et al. [25].

The S1 (stickiness of surface layer) of chalky cooked rice grains (−0.004–−0.009 × 10^5^ (N/cm^2^); mean, −0.006 × 10^5^ (N/cm^2^)) were lower than those of overall cooked rice grains (−0.004–−0.011 × 10^5^ (N/cm^2^); mean, −0.007 × 10^5^ (N/cm^2^)). The S1 of chalky cooked rice grains were 0.9 times lower than those of overall cooked rice grains. The S2 (stickiness of overall layer) and L3 (the adhered of surface layer) of chalky cooked rice grains were almost the same as those of overall cooked rice grains.

The values for Balance degree, H1(S1/H1), H2(S2/H2), A1(A3/A1), and A2(A6/A4) are important indices in evaluating the palatability of rice [35]. The Balance degree, H1, H2, A1, and A2 of chalky cooked rice grains were 1.1~1.4 times higher than those of overall cooked rice grains. As a result, in all samples, the H2 of chalky cooked rice grains were significantly lower than those of overall cooked rice grains, and those of H1 were also lower than those of overall cooked rice grains.

Low-amylose rice cultivars were found to be stale resistant: many researchers have previously reported the staling characteristics of cooked low-amylose rice [49]. As a staling test for cooked rice, the cooked samples were stored at 6 °C for 24 h and subjected to the second measurement with a Tensipresser, as shown in Table 4B.

The H1(R) (retrograded hardness of surface layer) of chalky cooked rice grains (0.065–0.161 × 10^5^ (N/cm^2^); mean, 0.096 × 10^5^ (N/cm^2^)) were slightly lower than those of whole cooked rice grains (0.072–0.127 × 10^5^ (N/cm^2^); mean, 0.097 × 10^5^ (N/cm^2^)) and the H2 (R) (retrograded hardness of overall layer) of chalky cooked rice grains (1.513–2.115 × 10^5^ (N/cm^2^); mean, 1.827 × 10^5^ (N/cm^2^)) were lower than those of whole cooked rice grains (1.567–2.071 × 10^5^ (N/cm^2^); mean, 1.842 × 10^5^ (N/cm^2^)). Moreover, the S1 (retrograded stickiness of surface layer), S2 (retrograded stickiness of overall layer), L3 (retrograded adhesion of surface layer), and the retrograded Balance A1 of chalky cooked rice grains were 0.9 times lower than those of overall cooked rice grains. The retrograded Balance, H1, H2, and A2 of chalky cooked rice grains were almost the same as those of overall cooked rice grains.

Moreover, the H1(RD) (retrogradation of degree of hardness of surface layer) of chalky cooked rice grains (1.37–3.07; mean, 1.98) were higher than those of overall cooked rice grains (1.09–1.90; mean, 1.55); moreover, the H2(RD) (retrogradation degree of hardness degree of overall layer) of chalky cooked rice grains (1.05–1.38; mean, 1.21) were higher than those of overall cooked rice grains (0.99–1.25; mean, 1.12). The H1(RD) and H2(RD) of chalky cooked rice grains were 1.1–1.3 times higher than those of overall cooked rice grains; additionally, those of S1(RD) (retrogradation degree of the stickiness of surface layer) were almost the same. The retrogradation degree of S2(RD), L3(RD), Balance H1(RD), H2(RD), A1(RD), and A2(RD) were 0.7–0.9 times lower than those of overall cooked rice grains. Singh et al. [25] reported that the chalky grains show lower cooking values, such as cooking time and water uptake, and textural parameters, such as hardness and cohesiveness. We showed that chalky grains show higher degrees of retrogradation after cooking, which means they are harder and less sticky than overall grains in this report. As described before [7,8], consumers need low-retrogradation rice grains. Chalky grains are not suitable for commercial use and rice for processing in terms of retrogradation. As Umemoto et al. [50] showed, the variation in SSIIa (starch synthase IIa gene) affects the eating quality after the storage of cooked rice at 5 °C; high-amylose and high-SLC rice cultivars seem to retrograde more markedly than low-amylose rice cultivars.

### 3.5. α- and β-Amylase Activities

Hakata et al. [51] showed that in developing seeds, high temperature induced the expression of α-amylase genes and that suppression is a potential strategy for ameliorating grain damage from global warming. The α-glucosidase enzyme is predominantly localized in the inner endosperm, whereas α-amylase is localized mainly in the outer layers [52].

As shown in Table 5, the α-amylase activities of the chalky rice grains (0.011–0.025 Ug^−1^; mean = 0.018 Ug^−1^) were significantly higher than those of the whole rice grains (0.008–0.013 Ug^−1^; mean = 0.010 Ug^−1^) at *p* < 0.01. The α-amylase activities of chalky rice grains were 1.7 times higher than those of whole rice grains. Similarly, the β-amylase activities of the chalky rice grains (0.020–0.036 Ug^−1^; mean = 0.031 Ug^−1^) were significantly higher than those of the whole rice grains (0.018–0.026 Ug^−1^; mean = 0.021 Ug^−1^) at *p* < 0.01. The β-amylase activities of chalky rice grains were 1.5 times higher than those of whole rice grains. The α-amylase activities of the chalky rice grains of Sasanishiki (2.4 times), Hanaechizen (2.1 times), Morinokumasan (1.9 times), Koshihikari (Toukamachi) (1.5 times), Koshihikari (Shibata) (1.4 times), Koshihikari (Tochio) (1.3 times), and Koshihikari (Oguni) (1.2 times) were higher than those of whole rice grains, and those of the β-amylase activities showed a similar tendency. Kaplan and Guy et al. [53] showed that β-amylase induction and resultant maltose accumulation may function as a compatible-solute stabilizing factor in the chloroplast stroma in response to acute temperature stress. The α-amylase showed a positive correlation with β-amylase (r = 0.70; *p* < 0.01), D-glucose (r = 0.64; *p* < 0.05), and sucrose (r = 0.83; *p* < 0.01). The β-amylase showed a positive correlation with D-glucose (r = 0.55; *p* < 0.05) and sucrose (r = 0.79; *p* < 0.01).

As a result, it seems that the starch-degrading enzyme activities of Sasanishiki, Hanaechizen, Morinokumasan, Koshihikari (Toukamachi), and Koshihikari (Shibata) samples were affected by high-temperature ripening. The chalky rice grains showed significantly higher α-amylase and β-amylase activities than those of whole rice grains at *p* < 0.01. Mitsui et al. [14] reported that α-amylase mainly causes chalkiness in the high-temperature-damaged rice grains. Our results are consistent with their results and reported that not only α-amylase but also β-amylase activities were higher for chalky rice grains in this paper. Appendix A shows the relationship between the ratios of λ_max_/AAC and α-amylase activities of 36 *Japonica* rice samples (low-amylose rice cultivars were excluded from the 40 *Japonica* rice samples). As a result, this equation showed a correlation coefficient of 0.51, which showed that this equation can be applied to unknown rice samples. The high ratio of chalky rice samples showed a high positive correlation with λ_max_/AAC ratios and α-amylase activities.

### 3.6. RS Contents

Resistant starch is a starch that escapes digestion in the small intestine and that may be fermented in the large intestine. Four main subtypes of resistant starch have been identified based on their source or structure [54,55]. In general, starches rich in amylose are naturally more resistant to digestion and more susceptible to retrograde, where the SLC in amylopectin behaves in a similar manner with amylose by restricting starch swelling [56].

As shown in Table 5, the RS of the chalky cooked rice grains (0.28–0.54%; mean = 0.38%) was slightly higher than that of whole cooked rice grains (0.26–0.49%; mean = 0.37%). In particular, the RS of the chalky cooked rice of Koshihikari (Toukamachi) (1.2 times) and Hanaechizen (1.1 times) were significantly higher than those of whole cooked rice. The RS of the chalky cooked rice of Koshihikari (Tochio) (1.0 times) and Sasanishiki (1.0 times) were almost the same as those of whole cooked rice. The RS of the chalky cooked rice of Koshihikari (Shibata) (0.8 times) and Koshihikari (Oguni) (0.9 times) were lower than those of whole cooked rice. The RS of the cooked rice showed a positive correlation with H1(R), H2(R), and H1(RD). In general, the long chain of amylopectin (DP > 10) can significantly increase the amount of RS [57].

### 3.7. Glucose and Sucrose Contents

As shown in Table 5, the D-glucose contents of the chalky cooked rice grains (0.076–0.089%; mean = 0.083%) were significantly higher than those of whole cooked rice grains (0.056–0.078%; mean = 0.066%) at *p* < 0.01. Similarly, the sucrose contents of the chalky cooked rice grains (0.348–0.432%; mean = 0.388%) were significantly higher than those of whole cooked rice grains (0.276–0.325%; mean = 0.296%) at *p* < 0.01. The D-glucose and the sucrose contents of chalky cooked rice grains were 1.3 times higher than those of whole cooked rice grains, respectively. The D-glucose showed a positive correlation with sucrose (r = 0.85; *p* < 0.01). The reason chalky grains contain more glucose and sucrose than whole grains is due to the higher activities of amylases and lower activities of starch synthesizing enzymes [14].

### 3.8. SDS-PAGE

The protein content in rice grains is influenced by weather conditions [58]. The higher the protein content, the harder and less sticky the rice upon cooking [59,60,61,62,63].

As shown in Figure 3, the intensities of the bands of 13 kDa prolamin reveal the prolamin ratios of 1, Koshihikari (Niigata; Toukamachi), Whole grain (29.7%); 2, Koshihikari (Niigata; Toukamachi), Chalky grain (31.8%); 3, Koshihikari (Niigata; Shibata), Whole grain (31.3%); 4, Koshihikari (Niigata; Shibata), Chalky grain (30.1%); 5, Koshihikari (Niigata; Tochio), Whole grain (32.2%); 6, Koshihikari (Niigata; Tochio), Chalky grain (28.9%); 7, Morinokumasan (Kumamoto), Whole grain (31.1%); 8, Morinokumasan (Kumamoto), Chalky grain (29.8%); 9, Hanaechizen (Fukui), Whole grain (26.2%); 10, Hanaechizen (Fukui), Chalky grain (25.1%); 11, Koshihikari (Niigata; Oguni), Whole grain (29.3%); 12, Koshihikari (Niigata; Oguni), Chalky grain (26.8%); 13, Sasanishiki (Miyagi), Whole grain (29.3%); 14, Sasanishiki (Miyagi), Chalky grain (28.4%). The 13 kDa prolamin ratios of chalky rice grains (25.1–31.8%; mean, 28.7%) were lower than those of whole rice grains (26.2–32.2%; mean, 29.9%). Yamakawa et al. [64] reported that, using SDS-PAGE and immunoblot analysis of storage proteins, the accumulation of 13 kDa prolamin decreased, which is consistent with the diminished expression of prolamin genes under elevated temperature. Protein is the second most abundant constituent of milled rice after starch [65]. It affects the physical properties of cooked rice grains. In the present paper, the 13 kDa prolamin showed a negative correlation with H1(R) and H1(RD), and those of chalky rice grains were lower than those of whole rice grains. It is well known that rice with high protein content and high ratio of the 13 kDa prolamin shows inferior palatability [2,3]. We report in this paper that the cooked rice grains of chalky rice become softer and non-sticky because they contain a lower amount of 13 kDa prolamin.

### 3.9. Correlation among the Physicochemical and Physical Properties of Whole and Chalky Rice Grains of 7 Rice Samples with a High Ratio of Chalky Rice Grains

The relative correlations among the biological, chemical, and physical parameters of the whole and chalky rice grains of seven rice samples with a high ratio of chalky rice grains are shown in Table 6. The λ_max_ showed a positive correlation with 13 kDa prolamin (r = 0.61; *p* < 0.05), and negative correlations with α-amylase (r = − 0.61; *p* < 0.05), D-glucose (r = − 0.53; *p* < 0.05), sucrose (r = − 0.69; *p* < 0.01), and Pt (r = − 0.79; *p* < 0.01).

The AAC is higher than the actual amylose contents because of the long chain amylopectin binding with iodine [4,5]. The AAC showed a positive correlation with H1 (R) (r = 0.64; *p* < 0.05), H2(R) (r = 0.65; *p* < 0.05), H1(RD) (r = 0.59; *p* < 0.05), Max. vis. (r = 0.62; *p* < 0.05), Mini. vis. (r = 0.68; *p* < 0.01), Fin. vis. (r = 0.74; *p* < 0.01), and RS (r = 0.81; *p* < 0.01), and negative correlations with 13 kDa prolamin (r = − 0.72; *p* < 0.01); moreover, A_λmax_ and Fb_3_/Fa ratios showed a similar tendency with AAC.

The H1 showed negative correlations with D-glucose (r = − 0.56; *p* < 0.05) and the H2 showed negative correlations with β-amylase (r = − 0.63; *p* < 0.05).

The H1(R) showed a positive correlation with H2(R) (r = 0.84; *p* < 0.01), H1(RD) (r = 0.81; *p* < 0.01), H2(RD) (r = 0.59; *p* < 0.05), Fin. vis. (r = 0.53; *p* − 0.05), and RS (r = 0.69; *p* < 0.01), and negative correlations with 13 kDa prolamin (r = − 0.66; *p* < 0.05); furthermore, the H2(R) showed a similar tendency as H1(R).

The H1(RD) showed a positive correlation with H2(RD) (r = 0.69; *p* < 0.01), Fb_3_ (DP ≥ 37) (r = 0.64; *p* < 0.05), Fb_3_/Fa ratios (r = 0.68; *p* < 0.01), α-amylase (r = 0.54; *p* < 0.05), RS (r = 0.66; *p* < 0.05), and Pt (r = 0.55; *p* < 0.05), and negative correlations with 13 kDa prolamin (r = − 0.73; *p* < 0.01).

Similarly, H2(RD) showed a positive correlation with α-amylase (r = 0.71; *p* < 0.01) and Pt (r = 0.53; *p* < 0.05).

In a previous study [34], we found that pasting properties show correlation with the texture of cooked rice grains. Kitadume et al. [66] showed that protein content and α-amylase activity had a high negative correlation among whole and chalky rice grains.

In the present paper, the retrogradation and retrogradation degree of H1(R), H2(R), H1(RD) and H2(RD) show a negative correlation with 13 kDa prolamin; furthermore, the retrogradation degree of hardness H1(RD) and H2(RD) show a positive correlation with AAC, A_λmax_, Fb_3_/Fa ratios, α-amylase activities, and Pt. As a result, the retrogradation degree of hardness H1(RD) and H2(RD) showed a positive correlation with the super-long chain (SLC) contents of amylopectin.

### 3.10. Formula for Estimating the H1 (RD) of Cooked Whole and Chalky Rice Grains Based on the Pasting Properties Using RVA and α-Amylase Activities

In our previous paper, we developed a novel estimation formula for the H1(R) of various kinds of cooked rice and those of cooked glutinous rice based on the program at 120 °C using an RVA [7]. Moreover, we developed a novel estimation formula for the Balance degree of the surface layer (A3/A1) based on the iodine absorption curve of milled rice [34].

Figure 4A shows the formula for estimating the H1 (RD) of cooked whole and chalky rice grains based on the pasting properties using RVA and α-amylase activities.

The equation had a correlation coefficient (r) of 0.81 based on the calibration. The following formula for estimating the H1(RD) was obtained using whole rice grains from seven *Japonica* rice samples and those chalky rice grains. 1, Morinokumasan (whole grains); 2, Morinokumasan (chalky grains); 3, Koshihikari (Shibata) (whole grains); 4, Koshihikari (Shibata) (chalky grains); 5, Koshihikari (Tochio) (whole grains); 6, Koshihikari (Tochio) (chalky grains); 7, Koshihikari (Toukamachi) (whole grains); 8, Koshihikari (Toukamachi) (chalky grains); 9, Hanaechizen (whole grains); 10, Hanaechizen (chalky grains); 11, Koshihikari (Oguni) (whole grains); 12, Koshihikari (Oguni) (chalky grains); 13, Sasanishiki (whole grains); 14, Sasanishiki (chalky grains).

The H1(RD) of the chalky cooked rice of Hanaechizen (1.7 times), Koshihikari (Oguni) (1.4 times), Koshihikari (Toukamachi) (1.3 times), Morinokumasan (1.3 times), Sasanishiki (1.2 times), and Koshihikari (Shibata) (1.1 times) were higher than those of whole cooked rice, whereas that of Koshihikari (Tochio) (0.9 times) was lower than that of whole cooked rice.
H1 (RD) = 78.94 × α-amylase activities + 0.01 × Max/Mini − 3.57(2)

Figure 4B shows that a correlation coefficient (r) 0.81 was obtained with the application of the abovementioned formula for the validation test using 15 unknown samples. 1, Tsuyahime (Shimane); 2, Sagabiyori (Saga); 3, Oidemai (Kagawa); 4, Hatsushimo (Gifu); 5, Akihonami (Kagoshima); 6, Nikomaru (Kouchi); 7, Hitomebore (Miyagi); 8, Yukiwakamaru (Yamagata); 9, Akitakomachi (Akita); 10, Seitennohekireki (Aomori); 11, Koshihikari (Hyogo); 12, Koshihikari (Nagano); 13, Tsuyahime (Yamagata); 14, Koshihikari (Niigata; Muikamachi); 15, Koshihikari (Niigata; Sado).

These rice samples are shown in the Materials description column, and were a mixture of whole and chalky rice grains. The high degree of chalky rice samples have a tendency of higher H1(RD) values. Thus, as shown in Figure 4B, the validation test showed that Equation (1) could be applied to the unknown samples.

This would enable us to estimate the retrogradation degree of cooked rice grains, one of the cooking properties of rice samples, using α-amylase activity and pasting properties as explanatory variables, and to evaluate the characteristic qualities of the mixture of whole grain and chalky rice samples.

## 4. Conclusions

(1)Chalky rice grains showed lower values of pasting viscosities, such as Max. vis. Final vis. and lower AAC and 13kDa prolamin than the whole grains.(2)On the contrary, they showed higher values of λmax/AAC ratios, α-amylase and β-amylase activities, and glucose and sucrose contents.(3)Cooked rice grains from the chalky grains showed lower hardness and higher retrogradation degree after cooking compared with the whole grains.(4)Particularly, rice cultivars Morinokumasan, Koshihikari (Toukamachi), and Sasanishiki were damaged markedly by ripening under high temperature.

Furthermore, low-retrogradation rice is suitable as commercial rice. We developed an estimation formula for the H1(RD) of the *Japonica* rice samples ripened under high temperature based on the α-amylase activity and pasting properties. The correlation coefficients were 0.84 for calibration and 0.81 for the validation test. This formula could be used to estimate the cooking properties of the *Japonica* rice samples ripened under high temperature.

## Figures and Tables

**Figure 1 foods-11-00097-f001:**
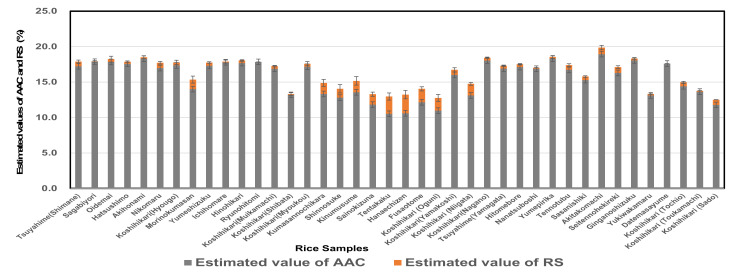
AAC and RS values of rice samples estimated based on pasting properties by an RVA; Estimated values of AAC (apparent amylose content) and RS (resistant starch) based on the pasting properties of 40 *Japonica* rice samples in 2019.

**Figure 2 foods-11-00097-f002:**
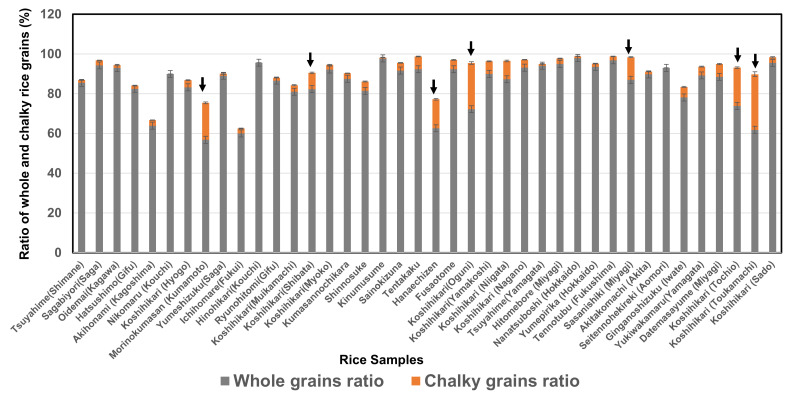
Ratio of whole and chalky rice grains of 40 *Japonica* rice samples in 2019.

**Figure 3 foods-11-00097-f003:**
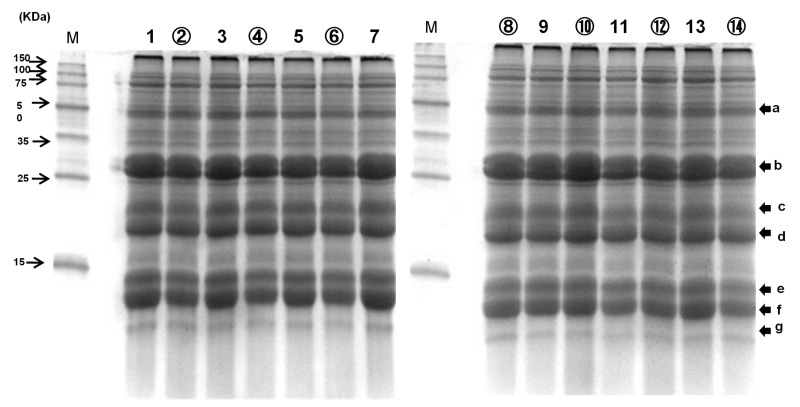
SDS-PAGE analysis of proteins extracted from whole and chalky rice grains of *Japonica* rice samples in a high temperature environment; 1, Koshihikari (Toukamachi) whole grain; 2, Koshihikari (Toukamachi) Chalky grain; 3, Koshihikari (Shibata) Whole grain; 4, Koshihikari (Shibata) Chalky grain; 5, Koshihikari (Tochio) Whole grain; 6, Koshihikari (Tochio) Chalky grain; 7, Morinokumasan (Whole grain); 8, Morinokumasan (Chalky grain); 9, Hanaechiden (Whole grain); 10, Hanaechzen (Chalky grain); 11, Koshihikari (Oguni) Whole grain; 12, Koshihikari (Oguni) Chalky grain; 13, Sasanishiki (Whole grain); 14, Sasanishiki (Chalky grain). SDS-PAGE; Sodium dodecyl sulfate polyacrylamide gel electrophoresis; Chalky rice grains is in circled numbers; a; Gluterin precursor, b; Glutelin α-subunit, c; α-Globulin, d; Glutelin β-subunit, e; f; g; Prolamin.

**Figure 4 foods-11-00097-f004:**
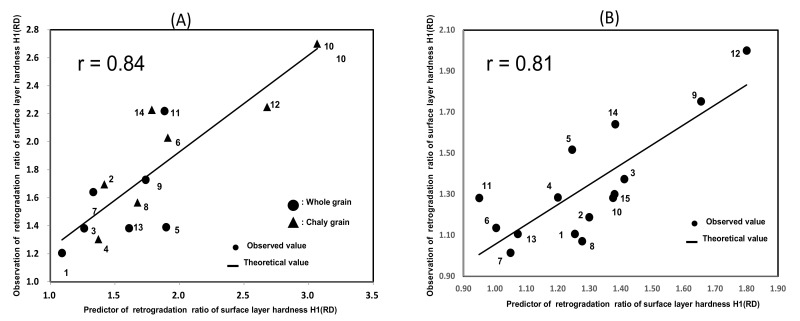
The estimation formula for the H1 (RD) of cooked rice grains based on the pasting properties using RVA and α- amylase activities; (**A**) Estimation formula: The following formula for estimating the H1(RD) was obtained using whole rice grains from 7 *Japonica* rice samples and those chalky rice grains. (1, Morinokumasan (whole grains); 2, Morinokumasan (chalky grains); 3, Koshihikari (Shibata) (whole grains); 4, Koshihikari (Shibata) (chalky grains); 5, Koshihikari (Tochio) (whole grains); 6, Koshihikari (Tochio) (chalky grains); 7, Koshihikari (Toukamachi) (whole grains); 8, Koshihikari (Toukamachi) (chalky grains); 9, Hanaechizen (whole grains); 10, Hanaechizen (chalky grains); 11, Koshihikari (Oguni) (whole grains); 12, Koshihikari (Oguni) (chalky grains); 13, Sasanishiki (whole grains); 14, Sasanishiki (chalky grains); (**B**) Examination estimation formula with unknown samples: The formula for validation test using 15 unknown samples (1, Tsuyahime (Shimane); 2, Sagabiyori (Saga); 3, Oidemai (Kagawa); 4, Hatsushimo (Gifu); 5, Akihonami (Kagoshima); 6, Nikomaru (Kouchi);7, Hitomebore (Miyagi); 8, Yukiwakamaru (Yamagata); 9, Akitakomachi (Akita); 10, Seitennohekireki (Aomori); 11, Koshihikari (Hyogo); 12, Koshihikari (Nagano); 13, Tsuyahime (Yamagata); 14, Koshihikari (Niigata; Muikamachi); 15, Koshihikari (Niigata; Sado); H1(RD); retrogradation ratio of surface layer hardness; The formula developed to estimate the H1(RD) based on the *Japonica* rice samples in a high temperature environment based on the α-amylase activities and pasting properties, of which the correlation coefficient was 0.84 for calibration, and 0.81 for validation test.

**Table 1 foods-11-00097-t001:** Pasting properties of *Japonica* polished rice by RVA in 2019.

	Max.vis.	Mini.vis.	BD	Fin.vis	SB	Pt	Cons	SB/	Max/	Max/
	(RVU)	(RVU)	(RVU)	(RVU)	(RVU)	(℃)	(RVU)	Cons	Mini	Fin
Tsuyahime (Shimane)	394.2 ± 7.0a	185.8 ± 1.2a	208.5 ± 5.8a	311.8 ± 0.6a	−82.4 ± 7.7a	63.9 ± 1.1a	126.0 ± 1.8a	−0.7 ± 0.1a	2.1 ± 0.0a	1.3 ± 0.0a
Sagabiyori	367.8 ± 11.6b	144.8 ± 0.1b	223.0 ± 11.7b	271.1 ± 1.5b	−96.7 ± 13.1b	63.0 ± 8.2b	126.3 ± 1.4a	−0.8 ± 0.1a	2.5 ± 0.1a	1.4 ± 0.0a
Oidemai	344.0 ± 1.9b	152.8 ± 1.5b	191.2 ± 0.4a	275.5 ± 1.0b	−68.6 ± 0.9c	63.5 ± 0.1c	122.6 ± 0.5a	−0.6 ± 0.0a	2.3 ± 0.0a	1.2 ± 0.0a
Hatsushimo	353.9 ± 4.2b	146.6 ± 1.8b	207.3 ± 2.5a	271.0 ± 2.5b	−83.0 ± 1.7a	63.5 ± 0.1c	124.4 ± 0.8a	−0.7 ± 0.0a	2.4 ± 0.0a	1.3 ± 0.0a
Akihonami	337.4 ± 2.5b	154.5 ± 1.0b	182.8 ± 3.5c	286.9 ± 1.4b	−60.5 ± 3.9d	63.0 ± 2.7a	132.4 ± 0.4a	−0.4 ± 0.0b	2.2 ± 0.0a	1.2± 0.0a
Nikomaru	350.9 ± 3.5b	157.5 ± 1.0b	193.3 ± 4.5a	283.3 ± 0.4b	−67.6 ± 3.8c	64.6 ± 0.0a	125.7 ± 0.6a	−0.5 ± 0.0b	2.2 ± 0.0a	1.2 ± 0.0a
Koshihikari (Hyougo)	365.2 ± 1.3b	155.8 ± 8.0b	209.3 ± 9.3a	275.0± 9.3b	−90.1 ± 10.5b	63.6 ± 0.5a	119.2 ± 1.2a	−0.8 ± 0.1a	2.3 ± 0.1a	1.3 ± 0.0a
Morinokumasan	343.9 ± 0.5b	153.0 ± 2.2b	190.9 ± 1.8a	271.4 ± 2.9b	−72.5 ± 2.4a	68.0 ± 0.4a	118.4 ± 0.6a	−0.6 ± 0.0a	2.2 ± 0.0a	1.3 ± 0.0a
Yumeshizuku	380.5 ± 9.0a	169.1 ± 3.3a	211.4 ± 12.3a	290.2 ± 3.2a	−90.3 ± 12.2b	63.5 ± 0.4c	121.1 ± 0.1a	−0.7 ± 0.1a	2.3 ± 0.1a	1.3 ± 0.0a
Ichihomare	355.8 ± 1.5b	142.3 ± 2.2b	213.5 ± 3.7a	259.2 ± 2.0b	−96.6 ± 3.6b	63.0 ± 7.2a	116.9 ± 0.2a	−0.8 ± 0.0a	2.5 ± 0.0a	1.4 ± 0.0a
Hinohikari	310.0 ± 1.8c	145.5 ± 0.3b	164.4 ± 2.1c	275.3 ± 0.5b	−34.7 ± 2.3d	63.5 ± 0.2a	129.7 ± 0.2a	−0.3 ± 0.0b	2.1 ± 0.0a	1.1 ± 0.0a
Ryunohitomi	341.4 ± 1.1b	131.0 ± 3.8b	210.4 ± 4.9b	222.5 ± 6.2c	−118.9 ± 7.2b	60.0 ± 0.6a	91.5 ± 2.4b	−1.3 ± 0.1c	2.6 ± 0.1a	1.5 ± 0.0a
Koshihikari (Muikamachi)	390.7 ± 0.6a	159.8 ± 0.5b	230.8 ± 1.1d	280.5 ± 1.6b	−110.2 ± 2.2b	63.5 ± 0.1c	120.7 ± 1.2a	−0.9 ± 0.0c	2.4 ± 0.0a	1.4 ± 0.0a
Koshihikari (Shibata)	396.5 ± 0.1a	157.0 ± 0.8b	239.5 ± 0.9d	280.0 ± 0.2b	−116.5 ± 0.3e	68.0 ± 2.1a	123.0 ± 0.6a	−0.9 ± 0.0c	2.5 ± 0.0a	1.4 ± 0.0a
Koshihikari (Myoukou)	353.1 ± 3.6b	145.9 ± 0.6b	207.3 ± 2.9a	269.7 ± 1.4b	−83.4 ± 2.2a	64.0 ± 0.5a	123.8 ± 0.7a	−0.7 ± 0.0a	2.4 ± 0.0a	1.3 ± 0.0a
Kumasannochikara	401.5 ± 6.3a	143.7 ± 1.6b	257.8 ± 4.7d	276.1 ± 1.1b	−125.4 ± 5.2e	68.4 ± 0.3d	132.4 ± 0.5a	−0.9 ± 0.0c	2.8 ± 0.0b	1.5 ± 0.0a
Shinnosuke	419.2 ± 0.6a	144.7 ± 0.1b	274.5 ± 0.6e	257.7 ± 0.0b	−161.5 ± 0.6f	67.0 ± 0.1d	113.0 ± 0.1a	−1.4 ± 0.0c	2.9 ± 0.0b	1.6 ± 0.0b
Kinumusume	402.4 ± 2.5a	155.4 ± 1.2b	247.0 ± 3.7d	288.2 ± 1.8b	−114.2 ± 4.2b	68.0 ± 0.0d	132.8 ± 0.5a	−0.9 ± 0.0c	2.6 ± 0.0a	1.4 ± 0.0a
Sainokizuna	445.7 ± 0.1d	149.7 ± 0.2b	296.0 ± 0.2e	266.3 ± 0.1b	−179.4 ± 0.1f	67.9 ± 0.5d	116.6 ± 0.3a	−1.5 ± 0.0c	3.0 ± 0.0b	1.7 ± 0.0b
Tentakaku	434.9 ± 1.7d	170.0 ± 1.7a	264.8 ± 0.0e	296.7 ± 2.3e	−138.2 ± 0.6e	70.4 ± 0.2d	126.7 ± 0.6a	−1.1 ± 0.0c	2.6 ± 0.0a	1.5 ± 0.0a
Hanaechizen	454.8 ± 2.9d	183.9 ± 0.4a	270.9 ± 3.4e	323.0 ± 0.9a	−131.8 ± 3.9e	70.9 ± 0.0d	139.1 ± 0.5a	−0.9 ± 0.0c	2.5 ± 0.0a	1.4 ± 0.0a
Fusaotome	437.1 ± 3.1d	145.5 ± 0.2b	291.6 ± 2.9e	281.0 ± 0.2b	−156.2 ± 3.3f	69.8 ± 0.5d	135.4 ± 0.4a	−1.2 ± 0.0c	3.0 ± 0.0b	1.6 ± 0.0b
Koshihikari (Oguni)	428.2 ± 2.7d	148.2 ± 1.9b	280.0 ± 0.8e	261.2 ± 2.2b	−167.0 ± 0.5f	68.8 ± 0.0d	113.0 ± 0.3a	−1.5 ± 0.0c	2.9 ± 0.0b	1.6 ± 0.0b
Koshihikari (Yamakoshi)	384.1 ± 10.6a	151.6 ± 2.8b	232.5 ± 7.8b	287.6 ± 2.1b	−96.5 ± 8.5b	65.2 ± 0.1a	136.0 ± 0.6a	−0.7 ± 0.1a	2.5 ± 0.0a	1.3 ± 0.0a
Koshihikari (Niigata)	411.0 ± 4.1a	156.8 ± 1.3b	254.2 ± 2.8d	282.8 ± 0.9b	−128.3 ± 3.1e	68.0 ± 0.0d	125.9 ± 0.4a	−1.0 ± 0.0c	2.6 ± 0.0a	1.5 ± 0.0a
Koshihikari (Nagano)	359.7 ± 2.1b	165.8 ± 0.3b	193.9 ± 2.4a	301.7 ± 0.3a	−58.0 ± 2.4d	63.5 ± 1.6b	135.9 ± 0.0a	−0.4 ± 0.0b	2.2 ± 0.0a	1.2 ± 0.0a
Tsuyahime (Yamagata)	384.4 ± 1.5a	151.2 ± 0.1b	233.2 ± 1.5b	276.5 ± 0.6b	−107.9 ± 0.9b	64.0 ± 0.0a	125.3 ± 0.6a	−0.9 ± 0.0c	2.5 ± 0.0a	1.4 ± 0.0a
Hitomebore	348.3 ± 8.5b	144.3 ± 3.4b	204.0 ± 5.2a	272.6 ± 3.4b	−75.7 ± 5.2a	64.0 ± 0.4a	128.3 ± 0.0a	−0.6 ± 0.0a	2.4 ± 0.0a	1.3 ± 0.0a
Nanatsuboshi	354.3 ± 4.1b	133.2 ± 1.2b	221.1 ± 2.9b	238.3 ± 1.5c	−116.0 ± 2.6e	63.2 ± 0.9a	105.1 ± 0.4b	−1.1 ± 0.0c	2.7 ± 0.0b	1.5 ± 0.0a
Yumepirika	357.8 ± 1.4b	159.9 ± 8.8b	197.9 ± 7.5a	286.7 ± 7.7b	−71.1 ± 6.4a	63.0 ± 5.1b	126.8 ± 1.1a	−0.6 ± 0.0a	2.2 ± 0.1a	1.2 ± 0.0a
Tennotubu	364.9 ± 1.8b	157.4 ± 0.9b	207.5 ± 2.7a	292.8 ± 1.0a	−72.1 ± 2.8a	64.7 ± 0.0a	135.4 ± 0.1a	−0.5 ± 0.0b	2.3 ± 0.0a	1.4 ± 0.0a
Sasanishiki	356.1 ± 2.4b	143.5 ± 1.6b	212.6 ± 0.8b	260.8 ± 1.8b	−95.3 ± 0.5b	66.0 ± 0.5a	117.3 ± 0.2a	−0.8 ± 0.0a	2.5 ± 0.0a	1.1 ± 0.0a
Akitakomachi	357.9 ± 16.3b	198.6 ± 31.2a	159.3 ± 15.0c	334.4 ± 22.3a	−23.6 ± 6.1g	64.3 ± 0.4a	135.8 ± 8.9a	−0.2 ± 0.0d	1.8 ± 0.2a	1.3 ± 0.0a
Seitennohekireki	378.1 ± 3.1a	160.6 ± 2.0b	217.5 ± 5.1b	287.9 ± 2.1b	− 90.2± 5.2b	65.0 ± 0.6a	127.3 ± 0.1a	−0.7 ± 0.0a	2.4 ± 0.0a	1.2 ± 0.0a
Ginganoshizuku	350.0 ± 4.4b	156.7 ± 3.9b	193.2 ± 0.5a	290.5 ± 5.1b	−59.5 ± 0.7c	63.2± 0.1c	133.7 ± 1.2a	−0.4 ± 0.0b	2.2 ± 0.0a	1.9 ± 0.0b
Yukiwakamaru	377.6 ± 3.6a	112.5 ± 1.2c	265.2 ± 2.4e	198.7 ± 1.2c	−179.0 ± 2.4f	64.4 ± 0.6a	86.2 ± 0.1b	−2.1 ± 0.0e	3.4 ± 0.0b	1.3 ± 0.0a
Datemasayume	398.7 ± 2.1a	140.3 ± 1.8b	258.4 ± 0.4d	252.6 ± 2.1b	−146.1 ± 0.0f	62.0 ± 0.3a	112.3 ± 0.4a	−1.3 ± 0.0c	2.8 ± 0.0b	1.6 ± 0.0b
Koshihikari (Tochio)	395.5 ± 0.5a	151.5 ± 1.2b	244.0 ± 1.6d	268.0 ± 1.1b	−127.5 ± 1.6e	66.0 ± 1.0a	116.5 ± 0.1a	−1.1 ± 0.0c	2.6 ± 0.0b	1.5 ± 0.0a
Koshihikari (Toukamachi)	403.8 ± 0.7a	156.3 ± 2.2b	247.5 ± 1.5d	300.0 ± 2.6b	−103.8 ± 1.9e	68.0 ± 0.0a	143.7 ± 0.4a	−0.7 ± 0.0c	2.6 ± 0.0b	1.3 ± 0.0a
Koshihikari (Sado)	393,9 ± 0.8a	157.0 ± 0.6b	240.0 ± 0.2d	280.0 ± 0.5c	−113.9 ± 0.2f	67.5 ± 0.0a	123.0 ± 0.1a	−1.0 ± 0.0e	2.5 ± 0.0b	1.4 ± 0.0a

Within each measure (Max.vis, Mini.vis, etc.) in the same column, different letters (a, b, c, etc.) denote statistically significant differences. Abbreviation: RVA, Rapid visco analyzer; SB, Setback (Final. vis-Max. vis); BD, Break down (Max. vis-Mini. vis); Max. vis, Maximum viscosity; Mini. vis, Minimum viscosity; Pt, Pasting temperature; Cons, Consistency (Fin. vis-Mini. vis); Fin. vis, Final. viscosity. Values are mean ± standard deviation.

**Table 2 foods-11-00097-t002:** Analysis of iodine absorption parameters of whole and chalky rice grains from 7 *Japonica* rice samples in 2019.

	λmax	Aλmax	AAC	λmax/	λmax/	Fb3 (DP ≧ 37)	New	Fb3/
	(nm)		(%)	AAC	Aλmax	(%)	λmax	Fa
Morinokumasan (wG)	569.0 ± 0.0a	0.293 ± 0.00a	12.9 ± 0.2a	44.0 ± 0.5a	1945.4 ± 14.1a	12.3 ± 0.1a	0.28 ± 0.0a	0.6 ± 0.0a
Morinokumasan (CG)	567.5 ± 0.7a	0.263 ± 0.00b	10.7 ± 0.0b	53.0 ± 0.1b	2161.9 ± 3.1b	11.0 ± 0.0b	0.24 ± 0.0a	0.4 ± 0.0b
Koshihikari (Shibata) (WG)	578.5 ± 4.9a	0.269± 0.00a	11.9 ± 0.1a	48.6 ± 0.0a	2150.6 ± 18.4a	11.2 ± 0.0a	0.22± 0.0a	0.4± 0.0a
Koshihikari (Shibata) (CG)	566.0 ± 1.4b	0.279 ± 0.00a	11.7 ± 0.1a	48.3 ± 0.3a	2028.7 ± 5.2a	11.7 ± 0.1a	0.27 ± 0.0a	0.5 ± 0.0a
Koshihikari (Tochio) (WG)	574.5 ± 0.7a	0.300 ± 0.00a	13.7 ± 0.1a	42.1 ± 0.3a	1918.2 ± 11.2a	12.6 ± 0.1a	0.27 ± 0.0a	0.6 ± 0.0a
Koshihikari (Tochio) (CG)	572.5 ± 2.1a	0.297 ± 0.00a	13.4 ± 0.2a	42.8 ± 0.4a	1927.6 ± 2.0a	12.5 ± 0.1a	0.27 ± 0.0a	0.6 ± 0.0a
Koshihikari (Toukamachi) (WG)	573.0 ± 1.4a	0.299 ± 0.00a	13.3 ± 0.3a	42.9 ± 1.1a	1916.5 ± 22.9a	12.6 ± 0.1a	0.27 ± 0.0a	0.6 ± 0.0a
Koshihikari (Toukamachi) (CG)	570.5 ± 3.5a	0.282 ± 0.01a	12.1 ± 0.4b	47.2 ± 1.2b	2023.3 ± 28.1b	11.8 ± 0.3b	0.26 ± 0.0a	0.5 ± 0.0b
Hanaechizen (wG)	567.0 ± 0.0a	0.348 ± 0.00a	16.5 ± 0.2a	34.4 ± 0.3a	1631.7 ± 10.0a	14.8 ± 0.1a	0.37 ± 0.0a	0.8 ± 0.0a
Hanaechizen (CG)	563.0 ± 2.8a	0.340 ± 0.00a	15.6 ± 0.3b	36.1 ± 0.5a	1658.3 ± 4.9b	14.4 ± 0.0a	0.39 ± 0.0a	0.8 ± 0.0a
Koshihikari (Oguni) (wG)	566.5 ± 2.1a	0.338 ± 0.01a	15.8 ± 0.4a	36.0 ± 0.9a	1676.2 ± 21.8a	14.3 ± 0.3a	0.36 ± 0.0a	0.8 ± 0.0a
Koshihikari (Oguni)(CG)	566.0 ± 0.0a	0.337 ± 0.00a	15.7 ± 0.1a	36.1 ± 0.2a	1679.5 ± 7.0a	14.3 ± 0.1a	0.37 ± 0.0a	0.8 ± 0.0a
Sasanishiki (wG)	575.5 ± 0.7a	0.339 ± 0.00a	16.1 ± 0.1a	35.6 ± 0.2a	1697.8 ± 23.3a	14.4 ± 0.2a	0.31 ± 0.0a	0.7 ± 0.0a
Sasanishiki (CG)	570.5 ± 2.1a	0.323 ± 0.00b	15.2 ± 0.1b	37.4 ± 0.2b	1766.3 ± 6.6b	13.7 ± 0.0b	0.32 ± 0.0a	0.7 ± 0.0a

Within each measure (λmax, Aλmax, etc.) in the same column and difference between whole grains and chalky grains in each sample, different letters (a, b, c, etc.) denote statistically significant differences. Abbreviations: AAC, apparent amylose content; λmax, peak wavelength on iodine staining; A λmax, absorbance at λmax; Fb_3_, proportions of long chains in amylopectin (DP > 37) (%); Fa, proportions of short chains in amylopectin (DP < 12) (%); Values are mean ± standard deviation. WG, Whole grain; CG, Chalky grain.

**Table 3 foods-11-00097-t003:** Pasting properties of whole and chalky rice grains from 7 *Japonica* rice samples in 2019.

	Maxi.vis.	Mini.vis.	BD	Fin.vis	SB	Pt	Cons	SB/	Max/	Max/
	(RVU)	(RVU)	(RVU)	(RVU)	(RVU)	(℃)	(RVU)	Cons	Mini	Fin
Morinokumasan (wG)	352.4 ± 7.1a	152.8 ± 2.8a	199.6 ± 4.4a	269.9 ± 1.2a	−82.5 ± 6.0a	66.1 ± 1.0a	117.1 ± 1.6a	−0.7 ± 0.1a	2.3 ± 0.0a	1.3 ± 0.0a
Morinokumasan (CG)	320.9 ± 7.5b	132.4 ± 1.0b	188.5 ± 6.5a	238.4 ± 0.8b	−82.5 ± 6.7a	66.1 ± 0.1a	106.0 ± 0.2b	−0.8 ± 0.1a	2.4 ± 0.0a	1.3 ± 0.0a
Koshihikari (Shibata) (WG)	395.7 ± 3.1a	150.2 ± 2.0a	245.5 ± 1.1a	266.2 ± 1.9a	−129.5 ± 1.2a	65.0 ± 0.5a	116.0 ± 0.1a	−1.1 ± 0.0a	2.6 ± 0.0a	1.5 ± 0.0a
Koshihikari (Shibata) (CG)	364.7 ± 1.5b	144.9 ± 0.5b	219.8 ± 0.9b	257.3 ± 0.4b	−107.5 ± 1.1b	67.1 ± 0.4a	112.4 ± 0.2a	−1.0 ± 0.0a	2.5 ± 0.0a	1.4 ± 0.0a
Koshihikari (Tochio) (WG)	396.5 ± 3.7a	142.9 ± 0.3a	253.6 ± 3.4a	257.5 ± 0.2a	−138.9 ± 3.5a	64.6 ± 0.1a	114.7 ± 0.1a	−2.1 ± 0.1a	2.8 ± 0.0a	1.5 ± 0.0a
Koshihikari (Tochio) (CG)	393.9 ± 0.8a	135.7 ± 0.6b	258.2 ± 0.2a	244.4 ± 0.5b	−149.5 ± 0.2b	65.2 ± 0.0a	108.7 ± 0.1b	−2.3 ± 0.0a	2.9 ± 0.0a	1.6 ± 0.0a
Koshihikari (Toukamachi) (WG)	400.7 ± 3.7a	155.2 ± 4.5a	245.5 ± 8.2a	273.6 ± 4.9a	−127.1 ± 8.6a	65.0 ± 0.1a	118.5 ± 0.4a	−2.5 ± 0.2a	2.6 ± 0.1a	1.5 ± 0.1a
Koshihikari (Toukamachi) (CG)	371.4 ± 0.1b	146.0 ± 0.2b	225.5 ± 0.1b	260.1 ± 0.5b	−111.3 ± 0.4b	68.0 ± 0.2a	114.1 ± 0.3a	−1.7 ± 0.0b	2.5 ± 0.0a	1.4 ± 0.0a
Hanaechizen (wG)	446.0 ± 1.4a	176.5 ± 0.9a	269.5 ± 0.4a	316.8 ± 0.6a	−129.3 ± 0.7a	67.5 ± 0.1a	140.3 ± 0.3a	−0.9 ± 0.0a	2.5 ± 0.0a	1.4 ± 0.0a
Hanaechizen (CG)	392.5 ± 0.8b	162.3 ± 0.4b	230.3 ± 0.4b	293.9 ± 1.2b	−98.7 ± 0.5b	68.5 ± 0.0a	131.6 ± 0.8b	−0.7 ± 0.0a	2.4 ± 0.0a	1.3 ± 0.0a
Koshihikari (Oguni) (wG)	433.3 ± 0.2a	152.3 ± 1.9a	281.1 ± 2.1a	268.1 ± 1.6a	−165.2 ± 1.8a	66.0 ± 0.2a	115.9 ± 0.3a	−1.4 ± 0.0a	2.8 ± 0.0a	1.6 ± 0.0a
Koshihikari (Oguni)(CG)	420.7 ± 0.5b	144.8 ± 0.7b	275.9 ± 0.2a	259.0 ± 0.7a	−161.7 ± 0.2a	67.0 ± 0.0a	114.2 ± 0.0a	−1.4 ± 0.0a	2.9 ± 0.0a	1.6 ± 0.0a
Sasanishiki (wG)	382.2 ± 0.2a	162.1 ± 0.3a	2201. ± 0.5a	303.2 ± 0.1a	−79.0 ± 0.1a	65.0 ± 0.1a	141.0 ± 0.4a	−0.6 ± 0.0a	2.4 ± 0.0a	1.3 ± 0.0a
Sasanishiki (CG)	342.9 ± 0.1b	151.3 ± 0.4b	191.6 ± 0.3b	282.8 ± 0.7b	−60.2 ± 0.6b	66.4 ± 0.0a	131.5 ± 0.3b	−0.5 ± 0.0a	2.3 ± 0.0a	1.2 ± 0.0a

Within each measure (Max. vis, Mini. vis, etc.) in the same column and difference between whole grains and chalky grains in each sample, different letters (a, b, c, etc.) denote statistically significant differences. Abbreviation: SB, Setback (Final. Vis-Max. vis); BD, Break-down (Max. vis-Mini. vis); Max. vis, Maximum viscosity; Mini. vis, Minimum viscosity; Pt, Pasting temperature; Cons, Consistency (Fin. vis-Mini. vis); Fin. vis, Final. viscosity. WG, Whole grain; CG, Chalky grain. Values are mean ± standard deviation.

**Table 4 foods-11-00097-t004:** Physical properties of cooked whole and chalky rice grains from 7 *Japonica* rice samples in 2019.

**Group**	**Sample**	**Surface**	**Overall**	**Surface**	**Overall**	**Surface**	**Surface**	**Overall**	**Surface**	**Overall**
		**layer**		**layer**		**layer**	**layer**		**layer**	
		**Hardness**	**Hardness**	**Stickiness**	**Stickiness**	**Adhesion**	**Balance**	**Balance**	**Balance**	**Balance**
		**(H1)**	**(H2)**	**(S1)**	**(S2)**	**(L3)**	**Degree H1**	**Degree H2**	**Degree A1**	**Degree A2**
		**×10^5^[N/cm^2^]**	**×10^5^[N/cm^2^]**	**×10^5^[N/cm^2^]**	**×10^5^[N/cm^2^]**	**[mm]**	**(S1/H1)**	**(S2/H2)**	**(A3/A1)**	**(A6/A4)**
**(A)**	Morinokumasan (wG)	0.07 ± 0.02a	1.61 ± 0.3a	−0.01 ± 0.00a	−0.36 ± 0.1a	3.01 ± 0.5a	0.09 ± 0.1a	0.23 ± 0.05a	0.17 ± 0.1a	0.08 ± 0.0a
	Morinokumasan (CG)	0.07 ± 0.01a	1.47 ± 0.2b	0.00 ± 0.00a	−0.33 ± 0.0a	2.67 ± 0.9a	0.07 ± 0.0a	0.23 ± 0.05a	0.13 ± 0.1a	0.09 ± 0.0a
	Koshihikari (Shibata) (WG)	0.06 ± 0.01a	1.59 ± 0.2a	0.00 ± 0.00a	−0.32 ± 0.1a	2.78 ± 0.8a	0.06 ± 0.0a	0.20 ± 0.05a	0.13 ± 0.1a	0.08 ± 0.0a
	Koshihikari (Shibata) (CG)	0.05 ± 0.02b	1.44 ± 0.2b	0.00 ± 0.00a	−0.31 ± 0.0a	2.81 ± 0.8a	0.10 ± 0.1a	0.22 ± 0.03a	0.22 ± 0.2a	0.10 ± 0.0a
	Koshihikari (Tochio) (WG)	0.06 ± 0.01a	1.76 ± 0.1a	−0.01 ± 0.00a	−0.30 ± 0.0a	3.01 ± 0.5a	0.17 ± 0.1a	0.17 ± 0.03a	0.32 ± 0.2a	0.10 ± 0.0a
	Koshihikari (Tochio) (CG)	0.04 ± 0.02b	1.65 ± 0.1b	−0.01 ± 0.00a	−0.32 ± 0.0a	3.05 ± 0.3a	0.22 ± 0.1a	0.20 ± 0.03a	0.44 ± 0.1b	0.13 ± 0.0a
	Koshihikari (Toukamachi) (WG)	0.07 ± 0.01a	1.57 ± 0.2a	0.00 ± 0.00a	−0.30 ± 0.1a	2.72 ± 0.7a	0.06 ± 0.0a	0.19 ± 0.07a	0.11 ± 0.1a	0.07 ± 0.0a
	Koshihikari (Toukamachi) (CG)	0.05 ± 0.02b	1.41 ± 0.2b	0.00 ± 0.00a	−0.30 ± 0.1a	2.68 ± 0.9a	0.09 ± 0.1a	0.22 ± 0.04a	0.21 ± 0.1a	0.09 ± 0.0a
	Hanaechizen (wG)	0.07 ± 0.01a	1.81 ± 0.2a	−0.01 ± 0.00a	−0.35 ± 0.1a	2.82 ± 0.8a	0.10 ± 0.0a	0.20 ± 0.03a	0.17 ± 0.1a	0.07 ± 0.0a
	Hanaechizen (CG)	0.05 ± 0.02b	1.54 ± 0.3b	0.01 ± 0.00b	−0.32 ± 0.1a	2.71 ± 0.9a	0.09 ± 0.1b	0.21 ± 0.04a	0.18 ± 0.1a	0.08 ± 0.0a
	Koshihikari (Oguni) (wG)	0.05 ± 0.01a	1.49 ± 0.2a	−0.01 ± 0.01a	−0.29 ± 0.1a	2.63 ± 0.9a	0.14 ± 0.1a	0.19 ± 0.04a	0.28 ± 0.2a	0.11 ± 0.1a
	Koshihikari (Oguni)(CG)	0.04 ± 0.01b	1.62 ± 0.2b	−0.01 ± 0.00a	−0.31 ± 0.0a	2.86 ± 0.8a	0.22 ± 0.1b	0.19 ± 0.03a	0.43 ± 0.2b	0.12 ± 0.0a
	Sasanishiki (wG)	0.06 ± 0.02a	1.69 ± 0.3a	−0.01 ± 0.01a	−0.33 ± 0.1a	2.87 ± 0.7a	0.16 ± 0.1a	0.20 ± 0.03a	0.27 ± 0.2a	0.11 ± 0.0a
	Sasanishiki (CG)	0.04 ± 0.01b	1.49 ± 0.2b	−0.01 ± 0.00a	−0.30 ± 0.1a	3.06 ± 0.4a	0.16 ± 0.1a	0.20 ± 0.03a	0.37 ± 0.2b	0.11 ± 0.0a
	**Sample**	**Surface**	**Overall**	**Surface**	**Overall**	**Surface**	**Surface**	**Overall**	**Surface**	**Overall**
		**layer**		**layer**		**layer**	**layer**		**layer**	
		**Hardness**	**Hardness**	**Stickiness**	**Stickiness**	**Adhesion**	**Balance**	**Balance**	**Balance**	**Balance**
		**(H1)**	**(H2)**	**(S1)**	**(S2)**	**(L3)**	**Degree H1**	**Degree H2**	**Degree A1**	**Degree A2**
		**×10^5^[N/cm^2^]**	**×10^5^[N/cm^2^]**	**×10^5^[N/cm^2^]**	**×10^5^[N/cm^2^]**	**[mm]**	**(S1/H1)**	**(S2/H2)**	**(A3/A1)**	**(A6/A4)**
**(B)**	Morinokumasan (wG)	0.07 ± 0.02a	1.50 ± 0.2a	−0.01 ± 0.00a	−0.28 ± 0.1a	2.64 ± 0.9a	0.08 ± 0.1a	0.18 ± 0.1a	0.16 ± 0.1a	0.08 ± 0.0a
	Morinokumasan (CG)	0.09 ± 0.03b	1.80 ± 0.2a	0.00 ± 0.00a	−0.31 ± 0.1b	2.36 ± 1.2a	0.05 ± 0.0a	0.18 ± 0.0a	0.08 ± 0.1a	0.05 ± 0.0a
	Koshihikari(Shibata)(WG)	0.06 ± 0.04a	1.57 ± 0.4a	0.00 ± 0.00a	−0.22 ± 0.1a	2.20 ± 1.1a	0.02 ± 0.0a	0.15 ± 0.1a	0.06 ± 0.0a	0.05 ± 0.0a
	Koshihikari(Shibata)(CG)	0.07 ± 0.02a	1.51 ± 0.3a	0.00 ± 0.00a	−0.25 ± 0.1b	2.64 ± 0.9a	0.04 ± 0.0a	0.17 ± 0.1a	0.10 ± 0.1a	0.06 ± 0.0a
	Koshihikari(Tochio)(WG)	0.11 ± 0.04a	2.02 ± 0.2a	0.00 ± 0.00a	−0.21 ± 0.1a	2.77 ± 0.7a	0.05 ± 0.0a	0.11 ± 0.1a	0.09 ± 0.1a	0.04 ± 0.0a
	Koshihikari(Tochio)(CG)	0.07 ± 0.04b	1.75 ± 0.2b	0.00 ± 0.00a	−0.24 ± 0.1b	2.12 ± 1.2a	0.07 ± 0.0a	0.14 ± 0.1a	0.11 ± 0.1a	0.06 ± 0.0a
	Koshihikari(Toukamachi)(WG)	0.09 ± 0.03a	1.96 ± 0.2a	−0.01 ± 0.00a	−0.34 ± 0.1a	2.74 ± 0.9a	0.12 ± 0.1a	0.18 ± 0.0a	0.17 ± 0.1a	0.07 ± 0.0a
	Koshihikari(Toukamachi)(CG)	0.08 ± 0.05a	1.80 ± 0.4a	−0.01 ± 0.00a	−0.31 ± 0.1b	2.43 ± 1.0a	0.12 ± 0.1a	0.18 ± 0.1a	0.20 ± 0.2a	0.10 ± 0.1a
	Hanaechizen (wG)	0.13 ± 0.04a	2.07 ± 0.2a	−0.01 ± 0.00a	−0.25 ± 0.1a	2.72 ± 0.9a	0.05 ± 0.0a	0.13 ± 0.1a	0.07 ± 0.1a	0.04 ± 0.0a
	Hanaechizen (CG)	0.16 ± 0.07b	2.11 ± 0.3a	0.00 ± 0.00a	−0.17 ± 0.1b	2.57 ± 0.9a	0.03 ± 0.0a	0.09 ± 0.1a	0.04 ± 0.0a	0.03 ± 0.0a
	Koshihikari(Oguni) (wG)	0.10 ± 0.05a	1.82 ± 0.2a	0.00 ± 0.00a	−0.24 ± 0.1a	2.80 ± 0.9a	0.06 ± 0.0a	0.14 ± 0.1a	0.11 ± 0.1a	0.05 ± 0.a
	Koshihikari(Oguni)(CG)	0.12 ± 0.07a	1.97 ± 0.3a	0.00 ± 0.00a	−0.14 ± 0.1b	1.85 ± 1.1b	0.05 ± 0.0a	0.08 ± 0.1b	0.063 ± 0.1a	0.04 ± 0.0a
	Sasanishiki (wG)	0.10 ± 0.05a	1.86 ± 0.sa	0.00 ± 0.00a	−0.26 ± 0.1a	2.73 ± 0.87a	0.06 ± 0.1a	0.15 ± 0.1a	0.12 ± 0.1a	0.05 ± 0.0a
	Sasanishiki (CG)	0.08 ± 0.03a	1.84 ± 0.2a	−0.01 ± 0.00a	−0.27 ± 0.1a	2.90 ± 0.6a	0.06 ± 0.1a	0.15 ± 0.1a	0.13 ± 0.1a	0.05 ± 0.0a

(A) shows the data for the whole grains, and (B) shows those for the chalky grains. Within each measure (H1, H2, etc.) in the same column and whole grains and chalky grains in each sample, different letters (a, b, c, etc.) denote statistically significant differences. H1; the hardness of the surface layer, H2; hardness of the overall layer, S1; the stickiness of the surface layer, S2; the stickiness of the overall layer, L3; the adhesion of the surface layer, Balance H1 (=S1/H1); the ratio of stickiness to hardness of surface layer, Balance H2 (=S2/H2); the ratio of stickiness to hardness of the overall layer, Balance A1 (=A3/A1); the ratio of adhesive work to hardness work of the surface layer, Balance A2 (=A6/A4); the ratio of adhesive work to hardness work of overall layer. SD; standard deviation. WG, Whole grain; CG, Chalky grain. Values are mean ± standard deviation.

**Table 5 foods-11-00097-t005:** Starch-hydrolyzing enzyme activities, sugar contents and resistant starch contents of whole and chalky rice grains from 7 *Japonica* rice samples in 2019.

	α-Amylase	β-Amylase	D-Glucose	Sucrose	RS
	(Ug^−1^)	(Ug^−1^)	(%)	(%)	(%)
Morinokumasan (wG)	0.012 ± 0.00a	0.025 ± 0.00a	0.077 ± 0.00a	0.325 ± 0.01a	0.26 ± 0.00a
Morinokumasan (CG)	0.022 ± 0.00b	0.036 ± 0.00b	0.089 ± 0.00b	0.432 ± 0.02b	0.28 ± 0.00a
Koshihikari (Shibata) (WG)	0.008 ± 0.00a	0.021 ± 0.00a	0.060 ± 0.00a	0.276 ± 0.01a	0.37 ± 0.00a
Koshihikari (Shibata) (CG)	0.011 ± 0.00b	0.035 ± 0.00b	0.076 ± 0.00b	0.368 ± 0.02b	0.29 ± 0.02b
Koshihikari (Tochio) (WG)	0.008 ± 0.00a	0.018 ± 0.00a	0.078 ± 0.00a	0.294 ± 0.01a	0.33 ± 0.01a
Koshihikari (Tochio) (CG)	0.011 ± 0.00b	0.020 ± 0.00a	0.087 ± 0.00b	0.348 ± 0.02b	0.34 ± 0.01a
Koshihikari (Toukamachi) (WG)	0.011 ± 0.00a	0.016 ± 0.00a	0.061 ± 0.00a	0.279 ± 0.02a	0.27 ± 0.01a
Koshihikari (Toukamachi) (CG)	0.016 ± 0.00b	0.027 ± 0.00b	0.076 ± 0.00b	0.351 ± 0.02b	0.32 ± 0.01b
Hanaechizen (wG)	0.012 ± 0.00a	0.018 ± 0.00a	0.065 ± 0.00a	0.310 ± 0.01a	0.49 ± 0.02a
Hanaechizen (CG)	0.025 ± 0.00b	0.029 ± 0.00b	0.086 ± 0.00b	0.422 ± 0.01b	0.54 ± 0.01b
Koshihikari (Oguni) (wG)	0.013 ± 0.00a	0.025 ± 0.00a	0.068 ± 0.00a	0.302 ± 0.01a	0.43 ± 0.01a
Koshihikari (Oguni)(CG)	0.015 ± 0.00b	0.033 ± 0.00b	0.084 ± 0.00b	0.411 ± 0.01b	0.38 ± 0.01a
Sasanishiki (wG)	0.010 ± 0.00a	0.026 ± 0.00a	0.056 ± 0.00a	0.286 ± 0.01a	0.46 ± 0.01a
Sasanishiki (CG)	0.023 ± 0.00b	0.037 ± 0.00b	0.084 ± 0.00b	0.384 ± 0.01b	0.48 ± 0.01a

Within each measure (α-amylase, β-amylase, etc.) in the same column and difference between whole grains and chalky grains in each sample, different letters (a, b, c, etc.) denote statistically significant differences. Values are mean ± standard deviation. WG, Whole grain; CG, Chalky grain.

**Table 6 foods-11-00097-t006:** Correlation between whole and chalky rice grains with the results of physical parameters of cooked rice, iodine analysis, pasting properties, RS content, α-and β-amylase activities, sugar contents and prolamin ratio of 7 rice samples.

	λ_max_	A_λmax_	AAC	Fb_3_/Fa	H1	H2	S2/H2	H1(R)	H2(R)	H1(RD)	H2(RD)	Maxi.vis.	Mini.vis.	Fin.vis	Pt	SB/Cons	α-Amylase	β-Amylase	D-Glucose	Sucrose	13kDaPlolamin	RS
λ_max_	1.00																					
A_λmax_	−0.35	1.00																				
AAC	−0.23	0.99 **	1.00																			
Fb_3_/Fa	−0.44	0.99 **	0.97 **	1.00																		
H 1	0.24	−0.06	−0.05	−0.08	1.00																	
H 2	0.30	0.41	0.47	0.38	0.44	1.00																
S2/H2	−0.39	−0.45	−0.52	−0.40	0.06	−0.61 *	1.00															
H 1(R)	−0.43	0.69 **	0.64*	0.72 **	0.14	0.36	−0.28	1.00														
(H2) R	−0.27	0.67 **	0.65*	0.68 **	0.07	0.43	−0.53	0.84 **	1.00													
H1(RD)	−0.53	0.64 *	0.59*	0.68 **	−0.46	0.05	−0.27	0.81 **	0.70 **	1.00												
H2(RD)	−0.50	0.37	0.30	0.41	−0.27	−0.33	−0.07	0.59 *	0.71 **	0.69 **	1.00											
Maxi.vis	−0.05	0.61 *	0.62 *	0.62 *	0.09	0.52	−0.64 *	0.48	0.46	0.35	0.06	1.00										
Mini.vis	−0.12	0.68 **	0.68 **	0.67 **	0.44	0.42	−0.16	0.52	0.42	0.19	0.11	0.51	1.00									
Fin.vis	−0.07	0.72 **	0.74 **	0.70 **	0.37	0.43	−0.19	0.53 *	0.46	0.23	0.14	0.40	0.97 **	1.00								
Pt	−0.79 **	0.27	0.17	0.33	−0.30	−0.36	0.46	0.42	0.24	0.55 *	0.53 *	0.03	0.29	0.26	1.00							
SB/Cons	−0.30	0.20	0.18	0.21	0.15	−0.14	0.51	0.16	−0.17	0.07	−0.06	−0.37	0.34	0.43	−0.15	1.00						
α-Amylase	−061 *	0.16	0.10	0.22	−0.32	−0.50	0.41	0.38	0.31	0.54 *	0.71 **	−0.42	−0.05	0.03	−0.28	0.46	1.00					
β-Amylase	−0.50	−0.06	−0.12	−0.02	−0.46	−0.63 *	0.55 *	−0.06	−0.23	0.24	0.24	−0.58 *	−0.31	−0.21	0.01	0.63*	0.70 **	1.00				
D-Glucose	−0.53*	−0.15	−0.21	−0.09	−0.56 *	−0.29	0.29	0.07	0.08	0.41	0.31	−0.43	−0.57 *	−0.51	0.27	0.05	0.64 *	0.55 *	1.00			
Sucrose	−0.69 **	−0.02	−0.10	0.05	−0.47	−0.44	0.48	0.25	0.13	0.53	0.48	−0.43	−0.35	−0.28	0.02	0.35	0.83 **	0.79 **	0.85 **	1.00		
13-kDa Plolamin	0.61 *	−0.76 **	−0.72 **	−0.80**	0.23	−0.16	−0.03	−0.66 *	−0.49	−0.73 **	−0.38	−041	−0.51 *	−0.59 *	0.62 *	−0.34	−0.44	−0.18	−0.20	−0.40	1.00	
RS	−0.21	0.80 **	0.81 **	0.80 **	−0.12	0.24	−0.30	0.69 **	0.55 *	0.66 *	0.39	0.40	0.64 *	0.74 **	−072 **	0.44	0.38	0.09	-0.09	0.10	−0.74 **	1.00

Correlation is significant at 5% (*) or 1% (**) by the method of *t*-test. H1(R); retrogradation of the hardness of surface layer, H2(R); retrogradation of the hardness of overall layer, H2(R.D); retrogradation degree of the hardness of overall layer, S2(R.D); retrogradation degree of the stickiness of the overall layer, Aλmax; absorbance at the peak wavelength (λmax). Retrogradation degree of hardness H1(RD) and H2(RD) showed positive correlation with AAC, λmax, Aλmax, Fb3/Fa ratios and α-amylase. As a result, the retrogradation degree of hardness H1 and H2(RD) showed a positive correlation with super-long chain (SLC) contents of amylopectin.

## Data Availability

The datasets generated for this study are available on request to the corresponding author.

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
