# Peer review of "Characteristics of Physicochemical Properties of Chalky Grains of Japonica Rice Generated by High Temperature during Ripening"

_foods, 2021, doi:10.3390/foods11010097_

Round 1

Reviewer 1 Report

Characteristics of Physicochemical Properties of Chalky Grains  of Japonica Rice Generated by High-temperature during Ripening.

  1. The abstract does not contain the purpose of the work and no sense of research.
  2. The introduction is haphazardly written. This should start with information about rice as a food crop, and only then should the characteristics of changes in rice grains occur. The introduction is to familiarize the reader with the topic of the work. It should not give the impression of a discussion of the results - this is the case here. The introduction requires major changes.
  3. A large number of sources were used in the introduction. It seems that the number of these sources could be smaller and the content more focused on introducing the reader to the research topic. Many sources are more than 10 years old. This needs to be changed.
  4. The reference list should be adapted to the requirements of the journal, e.g. lines 31, 34, 35, 62 etc.
  5. How do you know under what conditions the purchased rice was grown? How was this verified?
  6. “Milled rice  grains”  - how ground?
  7. “Okabe showed  that  the  rice  cooking  method  using  a  cup  of  measure  the  physical properties [38].” - This is a sentence for a discussion of results, not a methodology.
  8. There is no information on how to store the samples.
  9. The methodology for electrophoresis should be written in more detail.
  10. Tables and charts in the text are illegible. Perhaps the results should be presented differently.
  11. “Pasting properties are useful quality indicators because they affect the eating quality of rice [28,52]” - Is it really worth citing such a literature source? Horiuchi, H. Correlation among the amylograph characteristics of rice starch and flour. Agric. Biol. Chem. 1967, 31, 1003-1009.
  12. For the conclusions to be useful, the conditions under which rice is grown should be known.

Author Response

To Reviewer 1

     We are grateful for your precious reviewing and comments. As we revised our manuscript according to the comments, we appreciate very much if you kindly review it again.

  1. The abstract does not contain the purpose of the work and no sense of research.

We revised our abstract and we clarified the objectives of our investigation in L10-11.

  1. The introduction is haphazardly written. This should start with information about rice as a food crop, and only then should the characteristics of changes in rice grains occur. The introduction is to familiarize the reader with the topic of the work. It should not give the impression of a discussion of the results - this is the case here. The introduction requires major changes.

     We revised our Introduction. We added the information about rice in L27-40. Then we described the change in rice quality after L41. Furthermore, we re-arranged and revised our Introduction. 

  1. A large number of sources were used in the introduction. It seems that the number of these sources could be smaller and the content more focused on introducing the reader to the research topic. Many sources are more than 10 years old. This needs to be changed.

     We decreased the number of references in Introduction (former [2], [3], and [10] )

We also deleted the old references such as reference [9].

  1. The reference list should be adapted to the requirements of the journal, e.g. lines 31, 34, 35, 62 etc.

     We revised the references [31], [34], [35], [62] to new [27], [30], [31], [59].

  1. How do you know under what conditions the purchased rice was grown? How was this verified?

     First of all, all the rice grown in 2019 in Japan were damaged by very hot summer. As you know well, there are many problems about the qualities of rice grains in the market. Recently, increase of chalky grains became a big problem in Japan because chalky grains cause low milling yield, high breakage during milling and low eating quality as we pointed out in the text. Although it is very important and meaningful to use the rice samples of which growing condition is known, we, post-harvest researcher, must use the rice samples in the market. Therefore, we selected high-chalky grain rice samples and fractionated to two groups, chalky grain group and whole grain group. We think also our results are meaningful to characterize the quality change of chalky grains.

  1. “Milled rice grains” - how ground?

     We purchased milled rice grains as samples in the market and ground to the flours using Udy cyclone mill as described in L115-118.

  1. “Okabe showed that the rice cooking method using a cup of measure the physical properties [38].” - This is a sentence for a discussion of results, not a methodology.

     We deleted the reference 38 by Okabe because it is very old reference in L151 and 152.

  1. There is no information on how to store the samples.

     We stored the rice samples in the refrigerator (about 10 °C ) after collecting before the measurements as shown in L107.

  1. The methodology for electrophoresis should be written in more detail.

     We revised methodology of SDS-PAGE in L180 to L186.

  1. Tables and charts in the text are illegible. Perhaps the results should be presented differently.

     As you pointed out the methods for Figure 1 and Table 2 are different. AACs of Figure 1 are values estimated by the formulae based on RVA values of total rice flour (contains both chalky grains and whole grains) and AAC values in Table 2 are actual values using starch from each group of chalky grains or whole grain group.

But, Figure 1 shows that low-amylose rice samples tend to contain higher ratio of resistant starch and high-amylose rice group contained lower ratio of resistant starch, which is against usual tendency that amylose content and resistant starch content have positive correlation. It is very interesting and we think that this new negative correlation is affected by the increase of chalky grains.    

  1. “Pasting properties are useful quality indicators because they affect the eating quality of rice [28,52]” - Is it really worth citing such a literature source? Horiuchi, H. Correlation among the amylograph characteristics of rice starch and flour. Agric. Biol. Chem. 1967, 31, 1003-1009.

     We deleted it.

  1. For the conclusions to be useful, the conditions under which rice is grown should be known.

     Same answer with No5.

Reviewer 2 Report

Overall this MS need to revised fits to 12-14 pages and need to improved by proofers

Tables and figures need to be improved. 

Detailed suggestion (abstract, Methods, Results and Conclusion)  see attached file

Author Response

Answer to reviewer 2

  We are grateful for your precious reviewing and comments. As we revised our manuscript according to the comments, we appreciate very much if you kindly review it again.

Abstract

  • Avoid to repeat the same word in a sentence
  • Refine the abstract
  • Note at p< 0.01 (mean or equal with highly significant) and p<0.05 (mean significant)

Abstract maximum 200 words. Please follow the author guidance and write concisely:

  • Short Background (Main problem or the importance of this work)
  • Objective: The study was done to evaluated chalkiness of 40 Japonica rice grains harvested in Japan in 2019.
  • Methods. Seven samples with a high ratio of chalky rice grains were selected and divided in two groups (the whole and chalky group). The observed parameters were:
  • Results showed that surface layer hardness of cooked rice of chalky grains was highly significant lower and overall hardness the whole grains was lower significantly than whole grains. In addition, chalky rice grains had a highly significant higher of α- amylase 18 activities, β-amylase activities, D-glucose contents and sucrose contents than the whole rice grains. The developed of estimation formulae …..
  • This results presents the formula that could be used to estimate and to characterize of the cooking properties of the rice samples ripened under high temperature

Answer: Thank you very much for your precious and kind comments. According to your comments, we revised abstract from L9-L22.

Introduction Need to be refined

A: We revised our Introduction from L27 to L89.

Material & Methods need to be refined

Explain what RS (2.7 line 143)

A: We added the explanation on RS from L146 to L148 and L435 to L437.

A: We revised Table 1

Results Pay attention in presenting Table size or Figure. Please follow the guidance

Table 1; Since the mean value follow by a letter of significant different. You may remove the SD to add more space

Figure 1 and Figure 2: Presenting in Table form to make it more clear

Table 2, Table 3, Table 4 and Table 5: Improve the Table performance by using abbreviation and remove the SD (M-WG = Morinokumasan-whole grain), etc.

(0.011-0.025 385 U g-1; mean = 0.018 U g-1)

Improve the quality of Figure 3 Follow the guidance

A: We revised Table1-6 and Figure1 and 2 according to the comments.

Pay attention on international units

Please use the international units

(0.011-0.025 385 U/g; mean = 0.018 U/g) , and etc.

A: We revised units in Table5. 

3.6. Line 467: Refine the sentence and paragraph

A: We revised in L.472.

Table 6. the font too small

A: We revised Table 6.

(A)Estimation formula (page 16)

Please pay attention on the font size: follow the guidance

The font size to small (below the figure of regression)

For what estimation formula?

A: We reduced the font size of estimation formula.

A: we added in L534 to L535 explaining that the formula for estimating the degree of retrogradation of surface layer hardness, H1 (RD).

Conclusion Need to be refined. You may delete or remove the first paragraph.

How do you conclude that global warming impairs grain filling in rice and leads to chalky-appearing grains, (line544) which damages their physicochemical qualities and their market values.

The next sentence (line 545 to 549) is belonged to material and methods (not a conclusion)

A: We deleted L569 to L575 according to the comments.

Change we developed a: The developed of estimation formula for the H1(RD) of the japonica rice samples ripened under high temperature based on the α- amylase activity and pasting properties (the correlation coefficient was 0.84, and 0.81 for validation test) could be used to estimate the cooking properties of the japonica rice samples ripened under high temperature

A: We revised the paragraph, L576 to L583, to L584 to L591.

A: We revise L592 to L597.

Thank you very much for your valuable comments.

Reviewer 3 Report

The authors studied the physicochemical properties and cooking properties of a set of rice genotypes with different grain appearance (chalky and not chalky). The authors ascribe the presence of chalky grains to the very high temperature during grain filling.

The manuscript is well organized and written. The topic and the data reported are relevant and well discussed. The manuscript covers a good scope, relevant to the Journal.

From my point of view the manuscript is suitable for publication after minor comments and modifications:

  • Paragraph 2.1- Materials: to associate the presence of chalky grains with high temperatures, it would be important to have weather data of grain ripening and harvesting period.
  • Lines 127-132: not clear, please check the text.
  • Line 137: check the bibliographic quote.
  • Lines 148-149: not clear, please check the text.
  • Lines 188-191: genotypes with 20% (Figure 1) of AAC are considered “high amylose rices”, but it is intermediate amylose content
  • Lines 204-209: It is not clear how the authors selected the chalky genotypes; please check the text.
  • Wouldn't it be clearer to use the term "intact grain" instead of "whole grain"? This term often refers to whole grain cereal, so it could cause confusion.
  • Lines 292-294: not clear, please check the text.
  • Line 247: Do you mean “divided”?

Author Response

Answer to Reviewer 3

Thank you very much for your valuable reviewing. As we revised our manuscript according to your comments, we appreciate very much if you kindly review our revised manuscript again.

  1. The authors studied the physicochemical properties and cooking properties of a set of rice genotypes with different grain appearance (chalky and not chalky). The authors ascribe the presence of chalky grains to the very high temperature during grain filling.

  The manuscript is well organized and written. The topic and the data reported are relevant and well discussed. The manuscript covers a good scope, relevant to the Journal.

From my point of view the manuscript is suitable for publication after minor comments and modifications:

Thank you very much for your kind comments.

  1. Paragraph 2.1- Materials: to associate the presence of chalky grains with high temperatures, it would be important to have weather data of grain ripening and harvesting period.

According to your comment, we added explanation about the weather data from L241 to L243, and add the data in Supplemental Figure 2.

  1. Lines 127-132: not clear, please check the text.

According to your comment, we revised the text to L148-L159.

  1. Line 137: check the bibliographic quote.

According to your comment, we revised the text to L164-L168.

  1. Lines 148-149: not clear, please check the text.

According to your comment, we revised the text from L172 to L175.

  1. Lines 188-191: genotypes with 20% (Figure 1) of AAC are considered “high amylose rices”, but it is intermediate amylose content

According to your comment, we replaced high-amylose rice to intermediate rice from L227 to L228.

  1. Lines 204-209: It is not clear how the authors selected the chalky genotypes; please check the text.

First, we selected 8 rice samples based on the ratio of chalky rice grains, but, we removed one rice sample (Koshihikari, Niigata, dotted sample in Supplemental Figure 1) because it had very high ratio of whole grains (87.3%), which means low damage by the high temperature condition as described in Supplemental Figure 1.

  1. Wouldn't it be clearer to use the term "intact grain" instead of "whole grain"? This term often refers to whole grain cereal, so it could cause confusion.

According to your comment, we replaced “whole” to “overall” in the text (L360, 363, 370, 372, 374, 377, 379, 380, 393, 394, 397, 400, 401, 404, and 408). By changing from “whole” to “overall” in the section of physical properties of the cooked rice grains, we can avoid confusion with the “whole grain (non-damaged grain)”.

  1. Lines 292-294: not clear, please check the text.

According to your comment, we added the explanation from L272 toL273.

12.Line 247: Do you mean “divided”?

Yes, we selected 7 high-chalky rice samples, and we divided (fractionated) to two groups, chalky grain group, and whole grain group. Then the 14 rice samples were subjected to the various kinds of physicochemical measurements, such as AAC, RS, amylase activities, sugar content, pasting properties, physical properties (before and after retrogradation). As a result, it was shown that the chalky grains tend to easily retrograded after cooking compared with whole grains even though the same rice samples (same cultivar and same location).

Round 2

Reviewer 1 Report

Dear Authors,

a few changes were made to the manuscript. Some of them contribute a lot to the publication. Still, I think the introduction is chaotic and the changes made are minor.

I still believe that the general statement that the hot summer has affected the harvest does not make it possible to judge the changes in the rice. Detailed growing conditions are, in my opinion, necessary.

The literature list still does not comply with the journal's requirements. 

Author Response

Answer to Reviewer 1

Thank you very much for your valuable reviewing. As we revised our manuscript according to your comments, we appreciate very much if you kindly review our revised manuscript again.

  1. a few changes were made to the manuscript. Some of them contribute a lot to the publication. Still, I think the introduction is chaotic and the changes made are minor.

We revised introduction as follows;

(1) We added the explanation about the rice quality from L33 toL38.

(2) We added the explanation about the physicochemical measurements from L39 to L44.

(3) We added the explanation about the effects of chalkiness to rice quality and grading from L61 to L64.

(4) We added the explanation about the problems caused by the chalky grains from L65-L68.

(5) We revised to clarify our objective of this research from L103 to L106.

  1. I still believe that the general statement that the hot summer has affected the harvest does not make it possible to judge the changes in the rice. Detailed growing conditions are, in my opinion, necessary.

According to your comment, we added the description for growing conditions from L241 to L243 in text, and added Supplemental figure 2.

  1. The literature list still does not comply with the journal's requirements.

We are sorry. We revised our literature list according to the journal’s requirements, such as Italic style for volume for all the literatures, L626-627, L630-631, L732-L733, and L762-765.

Round 3

Reviewer 1 Report

After the changes have been made, I recommend the article for publication.